# Genome-wide gene-environment interaction study uncovers 162 vitamin D status variants using a precise ambient UVB measure

Rasha Shraim [1,2,3], Maria Timofeeva [4,5,6], Cathy Wyse [7,8], Jos van Geffen [9], Michiel van Weele [9], Roman Romero-Ortuno [10,11,12], Lorna M. Lopez [7,8,13], Marcus E. Kleber [14], Stefan Pilz [15], Winfried März [14,16,17], Benjamin S. Fletcher [18], James F. Wilson [18,19], Evropi Theodoratou [20,21], Malcolm G. Dunlop [4,20], Ross McManus [2] & Lina Zgaga [1] ✉

Vitamin D status is influenced by genetic and environmental factors—primarily sun exposure. Using satellite weather data, we estimated an ambient UVB dose for each participant based on residential address and date of sampling. We conducted genome-wide tests in 338,977 UK Biobank White British participants, adjusted for age, sex, supplements, UVB dose, and 10 principal components to account for population structure. We applied three models to test for genetic effects: marginal only, main and interaction, and joint effects. We identified 307 variants associated with standardised log-transformed 25-hydroxyvitamin D (25OHD) concentration, 162 of which were not previously identified in GWAS. We identify an increase in SNP-heritability by increasing ambient UVB exposure quintiles ($h^2_{Q1} = 8.48\%$ *vs.* $h^2_{Q5} = 15.56\%$). Downstream annotation implicated genes in the 25OHD pathway, including the circadian regulator, *BMAL1*. This and further findings suggest that vitamin D status and circadian rhythm may be entangled and that vitamin D metabolites may have a role as mediators of seasonal physiological fluctuations, including metabolism, and in turn explain the established associations with lipid metabolism pathways.

Several studies have examined the genetic architecture of vitamin D status and over 140 genetic variants have been identified in genome-wide association studies (GWAS) to date[1–4]. However, more so than most traits, vitamin D status is strongly affected by a single environmental exposure—solar ultraviolet B radiation (UVB)[5]. In the UK Biobank cohort, a weighted genetic risk score using 134 SNPs explained 4.2% of the variance in 25-hydroxyvitamin D concentration (25OHD, the best marker of vitamin D status), whilst ambient UVB explained substantially more, 12.4%[6]. Interestingly, varying heritability estimates have been reported for 25OHD depending on season of testing[7–9], with the most recent twin study reporting heritability of 0.37 in the winter and 0.62 in the summer[10].

Together, these observations suggest that accounting for solar radiation is important in understanding the genetic determinants of vitamin D status. Environmental and genetic contributions can be additive, meaning that the marginal genetic effect may be detectable regardless of the environmental context. Alternatively, if a gene-environment (GxE) interaction is present, the environmental effect on the trait is modified by the genotype[11]. If the interaction is not modelled, unexplained variance in the outcome is increased, and the genetic association may not be detected. Thus, incorporating environmental exposure data in GWAS and testing for interactions can reveal new insights into the aetiology of vitamin D status, and ultimately inform personalised interventions[12–14].

Earlier GWAS and candidate gene vitamin D studies reported evidence of GxE and noted the significant challenges of integrating environmental data into genetic studies[15]. For instance, Revez et al.[2] conducted a genome-wide variance quantitative trait locus (vQTL) analysis and identified 25 independent variants, only 5 of which showed evidence of interaction with season (others were reported as candidates for GxE with other exposures), while Manousaki et al.[3] evaluated interaction with season only in the GWAS significant variants. Environmental exposures relevant for vitamin D status are variable, and exposures over longer periods of time matter. Yet, the prevailing approach is to only use the season of blood draw as a proxy of environmental exposure. This is a rather imprecise, transient approximation of solar radiation, which varies substantially by day of the year, latitude, altitude, ozone, cloud cover and other factors. For example, in London on average a 50-fold difference was observed between annual high and annual low (average daily UVB dose was 0.11 kJ/m² in December and 5.56 kJ/m² in June), which illustrates the large variability[16].

Here, we aimed to advance the understanding of the genetic basis of vitamin D status, i.e. the concentration of 25OHD measured in the blood, by using a refined estimate of solar radiation relevant for vitamin D production. We calculated a unique cumulative and weighted ambient UVB dose based on each participant's residential location and date of blood draw for 25OHD measurement. We use this refined solar radiation measure in a genome-wide association and interaction study and report genome-wide marginal and interaction association effects using the UK Biobank, a large population-based cohort.

## Results

### 25OHD phenotype

After genetic quality control (QC) and exclusion of participants with no 25OHD or address data, 338,977 UK Biobank participants of White British ancestry (mean age: 56.9 y, 52.3% female) were included (Fig. 1, 'Methods'). 8,148,177 variants passed QC (Fig. S1). Genetic association analyses were adjusted for age, sex, vitamin D and fish oil supplements, 10 principal components (PCs), and cumulative and weighted UVB dose (CW-D-UVB) as the interacting environmental exposure with standardised log-transformed 25OHD as the outcome (Figs. S2,S3). Based on previous research from our group, a CW-D-UVB dose was calculated for each individual, using the Tropospheric Emission Monitoring Internet Service (TEMIS) database[17]. Briefly, CW-D-UVB is a cumulative measure of ambient UVB at a participant's place of residence over 135 days prior to blood draw, where daily doses are weighted so that the more recent UVB exposures contribute more—this reflects vitamin D accumulation and utilisation in the body ('Methods', Fig. S4).

Using GEM1.5.2[18], we applied three models testing for marginal, interaction, and joint effects. At a genome-wide significance level of $p < 5 \times 10^{-8}$, we identified 953 variants with interaction effects, 11,509 variants with marginal effects, and 11,715 variants with joint effects (Figs. 2, S5). The joint test, as implemented in GEM, indicates an association of the variant with 25OHD that is due to marginal *and/or* interaction effects. To identify significant independent SNPs from the marginal test results, comparable to previous vitamin D status GWAS, we used the method implemented in COJO-GCTA using an LD reference subset of 20,000 randomly selected White British participants, and we identified 105 such SNPs ('COJO variants'), 15 of which were low frequency variants (MAF < 0.05)[19]. To identify significant independent SNPs from the interaction and joint tests, we used the FUMA online pipeline[20]. 20 variants were identified as lead independent significant variants at $r^2 < 0.1$ from the interaction test (2 of these were also COJO variants, and 3 low-frequency), and 238 variants from the joint test (50 of these were also COJO variants, five were lead interaction variants, and one was reported in both). Thus, we identified 307 unique independent variants, 93 of which were low frequency. The average effect

### DATA

**UK Biobank**

| | |
|---|---|
| Phenotype | Standardised log-transformed 25-hydroxyvitamin D |
| Genotype | Imputed genotype QC [*Plink2*] |
| Environmental exposure | Cumulative & weighted ambient UVB radiation dose |
| Covariates | Age, sex, supplement intake, principal components |

**White British participants**

### GENETIC ASSOCIATION

**Discovery**: Genome-wide marginal only, main and interaction, and joint effects tests [*GEM 1.5*] (N = 338,977)

**Replication**: European UKB (N = 21,875), LURIC (N = 2,909), ORCADES (N = 1,875)

**Stratified analysis**
1) by UVB dose quintile (N = 74,398 each)
2) by BMI category ($N_{normal}$ = 120,925, $N_{overweight}$ = 159,671, $N_{obese}$ = 91,997)
3) in participants spending ≥3h/day outdoors (N = 172,273)

Independent significant variant selection [marginal *GCTA-COJO*, interaction & joint *FUMA*]

### DOWNSTREAM ANALYSIS

**Functional annotation:** Pathway & GO term enrichment [*FUMA, DAVID*]
SNP-based heritability, genetic correlation [*LDSC*]

**Fig. 1 | Study workflow.** Briefly, we used 25 hydroxyvitamin D (25OHD) as the outcome variable, covariates (age, sex, vitamin supplement use, fish oil use, and principal components), imputed genotype data from the UK Biobank (QC: quality control), and a unique ambient cumulative and weighted UVB (CW-D-UVB) dose, calculated for each individual based on their place of residence and date of blood draw. We performed genome-wide association and interaction analyses in the 'White British' group, and then conducted stratified genome-wide association and interaction analyses by BMI category (normal, overweight and obese), by quantile of ambient UVB, and for participants who spend 3 h or more outdoors per day. Results from the genetic analyses were further annotated for function and for SNP-based heritability.

estimate ($\beta_G$) for low frequency variants was −0.029 (s.d. 0.097), almost sixfold higher than that of common variants at −0.004 (s.d. 0.035). Similarly, for interaction, mean $\beta_{GxE}$ was −0.00016 (0.00042) and −1.9 × 10⁻⁶ (0.00013), respectively. After comparison to previously published results[1–4] and exclusion of previously reported variants or those in LD ($r^2 > 0.1$) with them, we report 162 additional variants associated with 25OHD, including 7 from the interaction test and 22 from the marginal test (Figs. 2, S5, Table S5).

### Replication

We used three samples for replication: the European cohort of the UKB ($N = 21,875$ European UKB participants excluded from the main analysis, mean age 55.7 y, 54.2% female), LURIC ($N = 2909$, mean age 62.9, 30% female; Germany)[21], and ORCADES ($N = 1875$, mean age 53.4 y, 60.4% female; Scotland)[22,23]. Of the 307 genome-wide significant variants from the discovery cohort, 305, 244 and 287 were available in the European, LURIC, and ORCADES samples, respectively. We compared the direction of the effect estimate in each cohort to the main White British cohort ($p_{binom}$ represents the $p$-value from a binomial test with a random sign of the effect estimate as the null hypothesis) as well as the correlation between the effect estimate in the main cohort and each of the replication cohorts (Fig. S6, Table S6). In the European replication, over 85% of the variants had the same sign across models (marginal and joint $p_{binom} < 2 \times 10^{-6}$, interaction $p_{binom} = 0.0026$). The

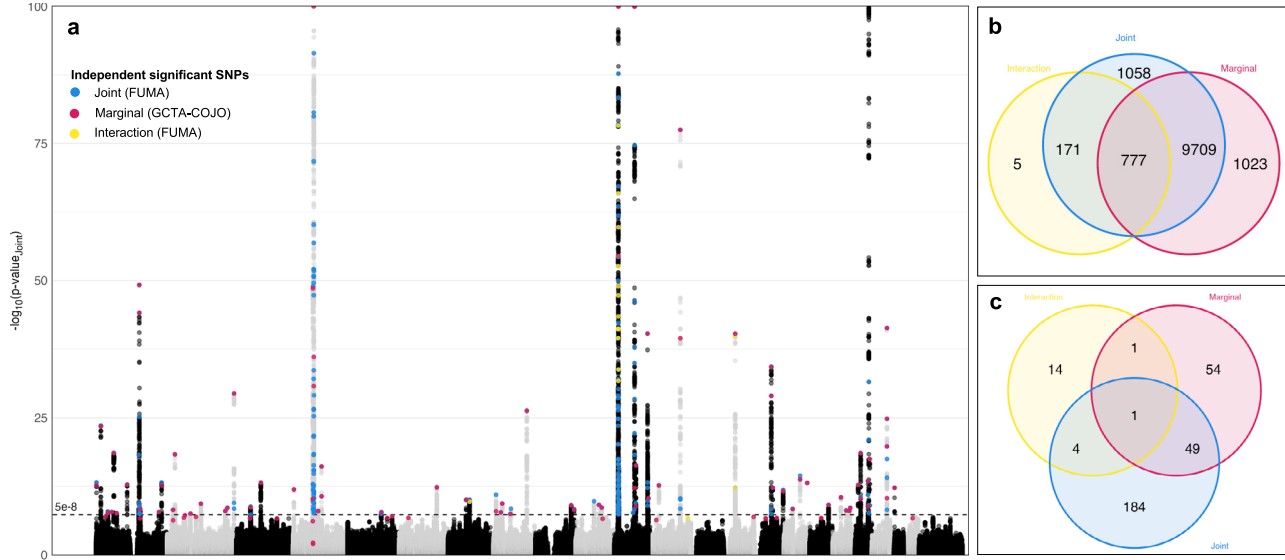

**Fig. 2 | Manhattan plot of the 25OHD genome-wide joint test in the UKB.**
**a** Manhattan plot of the genome-wide gene-environment interaction study of standardised, log-transformed 25OHD in the UK Biobank 'White British' population. The dots highlighted in blue, pink, and yellow represent the variants identified as genome-wide significant independent variants in the joint, marginal, and interaction test *p*-values, respectively. The *p*-values are shown on the −log10 scale and the dashed line shows the genome-wide adjusted significance cutoff line of *P* < 5 ×10⁻⁸, from the GEM joint test. Chromosome numbers are shown on the x-axis, where 23 represents the X-chromosome. Independent variants were selected using GCTA-COJO for the marginal test and FUMA for the joint and interaction test. The vertical axis is limited to 100 for readability. Overlap of marginal, interaction, and joint test results in: **b** all variants that reach genome-wide significance *p*-value < 5 × 10⁻⁸ in the present study (*N* = 12,743), and **c** independent significant variants selected by GCTA-COJO or FUMA (*N* = 307).

results from the binomial tests were also statistically significant for interaction in LURIC and for marginal and joint effects in ORCADES. Across the marginal variants, correlation with the discovery results was significant in all cohorts (correlation coefficient in European = 0.96, LURIC = 0.76, and ORCADES = 0.5). Interaction correlation was also significant in the European replication ($\beta_{GxUVB}$, correlation = 0.93). The observed weaker correlation in the interaction and joint test results, compared to the marginal test results, is likely partly mediated by the reduced power to detect GxE effects in these smaller samples, in addition to the reduced variance in the environmental exposure (lower UVB availability in the Orkney islands sample, Fig. S6c).

## SNP-based heritability

Previous studies demonstrate varying 25OHD heritability by season, with conflicting evidence on the direction of variation[2,3,7,8,24]. We used the marginal test summary statistics to estimate SNP-based heritability, i.e. the proportion of 25OHD variance explained by all SNPs. We first used LDSC and estimated 25OHD SNP-based heritability ($h^2_{SNP}$) to be 9.74% (s.e. 1.45, Fig. 3), based on the genetic effects of common variants. We also estimated the variance explained by the COJO independent SNPs with ≈ $2\beta^2 f(1-f)$, where $\beta$ is the marginal effect estimate and $f$ is the allele frequency: the total variance explained by the 105 conditionally independent SNPs was 5.24%. Low-frequency variants contributed 1.72% to $h^2_{SNP}$, which is less than common variants (3.52%), despite larger effect sizes (mean[$|\beta_{freq<0.05}|$] = 0.13; mean [$|\beta_{freq>0.05}|$] = 0.02). We then stratified our cohort according to CW-D-UVB quintiles (Fig. S7), and found that heritability increased with increasing ambient UVB dose, nearly doubling when comparing top and bottom quintiles (8.48% vs. 15.56%, Fig. 3). Similarly, $h^2_{SNP}$ was higher in a subset that spent ≥3 h outdoors (11.01%; s.e. 2.13, *N* = 172,273, Figs. 3, S8, S9). While the distribution of CW-D-UVB in this outdoors group was similar to that in the whole cohort, longer time outdoors likely led to a comparatively higher UVB dose received (Fig. S10). Our findings are comparable to previous 25OHD UKB GWAS of $h^2_{SNP}$ = 13% (s.e. 1.0) in ref. 2, and $h^2_{SNP}$ = 16.1%, and variance explained = 4.9% in ref. 3. Furthermore, $h^2_{SNP}$ estimates overall

increased when the sample was stratified by BMI (normal: 10.91%, overweight: 11.45% and obese: 10.78%, Fig. 3).

We additionally evaluated the genetic association of the significant independent variants with 25OHD in the replication cohort (*N* = 21,875, Fig. S11 shows the distribution of CW-D-UVB in the European group compared to White British group). Genetic risk scores were calculated based on the marginal effects ($\beta_{marginal}$) and on the main and interaction effects ($\beta_G$ + CW-D-UVB·$\beta_{GxE}$) estimated in the White British group. In an age and sex adjusted model, both scores were significantly associated with 25OHD in the European group ($p < 10^{-100}$, Table S4). 25OHD increased by 14.319 nmol/L (s.e. 0.485) for every unit increase in the marginal score (mean 0.43, s.d. 0.28) and by 5.21 nmol/L (s.e. 0.214) for the interaction score (mean 0.262, s.d. 0.647; Fig. S12 shows the distribution of the genetic scores).

## Functional annotation

Variants identified in each of the models were gene-annotated with Ensembl VEP. Of the 20 lead variants that were significant on the GxE interaction test, the direction of the effect estimate of interaction was the opposite of that of the main effect for only one variant (12:rs11058236). The majority of these variants were located in known 25OHD genes such as *CYP2R1*, *PDE3B*, *PSMA1*, *COPB1*, and *CALCB*. Three variants (rs11058236, rs12147280, on chromosomes 12 and 14, respectively, and variant 7:75653944) were not significant on the marginal test.

Further genes identified through significant marginal effects included among others, *PTH*, encoding the parathyroid hormone, that regulates blood calcium and phosphate levels, *LCE5A* known to be involved in keratinisation, *JMJD1C* that encodes a putative histone demethylase that interacts with thyroid hormone receptors, and *BMAL1/ARNTL*, encoding a transcription factor that serves as a core constituent in the transcription/translation feedback loop of the biological clock.

We then annotated the results from the GEM joint test with the FUMA online pipeline[20]. Gene-set analysis of the summary statistics identified 38 gene-sets or GO terms significantly associated with 25OHD (Fig. 4a), after Bonferroni correction. Enriched gene-sets were

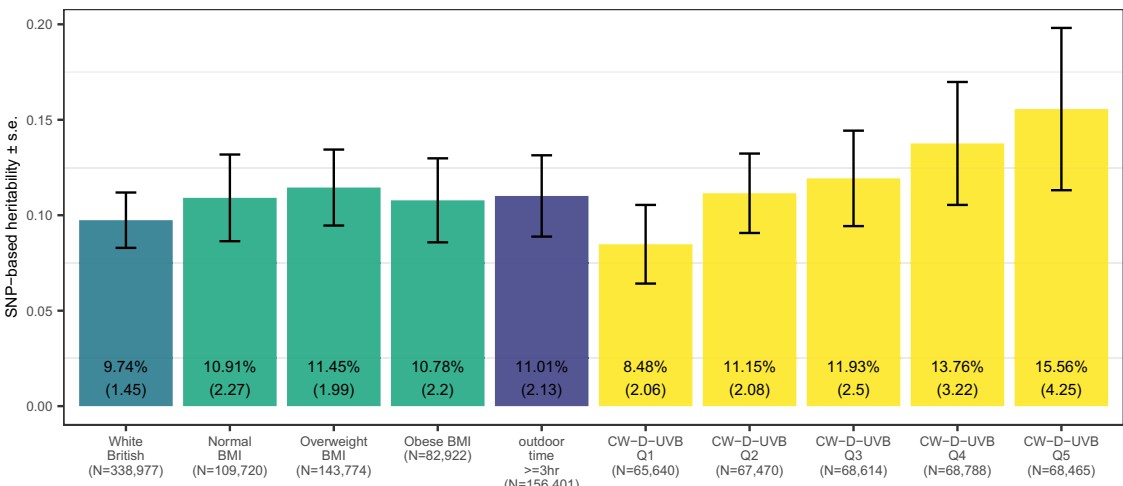

**Fig. 3 | SNP-based heritability in the overall cohort and stratified by subgroups.** SNP-based heritability [h²$_{SNP}$ (standard error)] was calculated based on the results of the marginal test for the entire cohort and within each subgroup: for participants who indicated spending 3 or more hours/day outdoors, by BMI category, and by CW-D-UVB quintiles. SNP-heritability estimates are presented with standard error bars.

dominated by different aspects of lipid metabolism. The top 5 terms were related to statin inhibition of cholesterol production, metabolic pathway of LDL, HDL and triglycerides (TG) including diseases, triglyceride-rich lipoprotein remodelling, chylomicron clearance, and acylglycerol homoeostasis. Other terms related to processes including glucuronidation and vitamin metabolism. Variants were primarily enriched for intronic and non-coding RNA splicing consequences (Fig. 4b, Table S13). In the MAGMA tissue expression analysis based on GTEx 58 tissue types, liver was the only tissue for which the adjusted p-value was significant (Figs. 4c, S13). Variants were also mapped to protein-coding genes and 194 protein-coding genes were identified as significantly associated with 25OHD. For example, the top gene was keratin-related, *KRTAP5-7* on chromosome 11.

To delineate the potential effects of the gene-environment interactions, we also applied the FUMA pipeline to the results from the interaction test only, albeit the gene-set analysis was underpowered. The top associated sets included genes upregulated in keratinocytes after UVB irradiation and *HOXC6* target genes (implicated in skeletal, skin, and endocrine/exocrine gland phenotypes), while the top tissue was adrenal gland ($p < 0.05$, $p_{Bonferroni} = 1$).

We then annotated all of our 307 selected significant variants with Ensembl VEP, which mapped to 148 genes[25]. We compared this list to genes identified in previous vitamin D status GWAS (GWAS Catalog 'vitamin D measurement')[26]. We identified 28 genes that had not been previously linked to 25OHD (Table S5). Functional enrichment analysis of the 148 mapped genes in DAVID identified 14 KEGG pathways enriched for 25OHD genes (FDR < 0.05; Fig. S14a)[27]. The top KEGG pathways identified were retinol metabolism and ascorbate and aldarate metabolism; the latter corresponds to the KEGG pathway identified in the joint FUMA gene-set analysis. The top mapped GO terms were cellular glucuronidation and glucuronosyltransferase activity, in addition to several lipid or cholesterol metabolism related terms (Fig. S14c). Disease annotation of these genes indicates enrichment of several lipid metabolism and cardiovascular outcomes (Fig. S14b).

### Genetic correlation

We evaluated the genetic correlation ($r_g$) of the marginal effects of 25OHD with several traits and health outcomes, selected primarily based on the traits considered in recent 25OHD GWA[2] and phenome-wide Mendelian randomisation[28] studies of vitamin D (Fig. 5, Table S14). Eight of the 25 selected phenotypes were significantly associated with 25OHD after Bonferroni correction for multiple testing, the majority of which were brain-related phenotypes. Among those, the most significant

genetic correlation was for intelligence ($r_g$= 0.231 (s.e. 0.030), $p = 1.09 \times 10^{-14}$), followed by autism spectrum disorder ($r_g = -0.225$ (0.047), $p = 1.99 \times 10^{-6}$), schizophrenia ($r_g = -0.108$ (0.023) $p = 3.66 \times 10^{-6}$), circadian rhythm ($r_g = 0.103$ (0.029), $p = 4 \times 10^{-4}$), and bipolar disorder ($r_g = -0.091$ (0.0275), $p = 9 \times 10^{-4}$). Additionally, BMI ($r_g = -0.136$ (0.026), $p = 1.5 \times 10^{-7}$), type 2 diabetes ($r_g = -0.176$ (0.037), $p = 2.08 \times 10^{-6}$) and melanoma ($r_g$=0.205 (0.049), $p = 2.91 \times 10^{-5}$) were also significantly associated with 25OHD. Phenotypes that did not reach the significance threshold included major depression disorder, anxiety disorder, blood pressure-related traits, Alzheimer's disease, and colorectal cancer (Fig. 5).

### BMI

BMI is a heritable trait and has been previously linked to vitamin D status, likely in a bidirectional manner[29]. Because of this and to avoid collider bias[30], we did not include BMI as a covariate, in line with previous GWAS[2,3]. Instead, we evaluated results from a stratified analysis by BMI category (normal $N = 120,925$, overweight $N = 159,671$, and obese $N = 91,997$; 'Methods'). The rationale for this is further supported by the abundance of gene-sets involved in lipid metabolism that were detected in the main analysis. The differences in the strength and distribution of associated SNPs suggest an additional interaction with BMI (Figs. S15, S16). In the joint test, the normal and overweight categories shared 63.5% of their genome-wide significant variants, normal and obese shared 54%, and overweight and obese shared 46%. For the GxUVB interaction test results, the overlap was 51%, 64%, and 61%, respectively. Although these differences may be influenced by the sample size of each category, this is not the case for variants that were significant in the normal and obese but not in the overweight group (which is the largest), thus supporting the presence of a BMI interaction. BMI-stratified functional annotation was also performed based on these results. The top 5 pathways, annotated in the MAGMA gene-set analysis, within each category were: in the normal BMI group, cellular and reactome glucuronidation, LDL/HDL/TG metabolism, chylomicron clearance, and hyperlipidemia ($p_{bon} < 0.05$); in the overweight group, statin inhibition of cholesterol production, LDL/HDL/TG metabolism, chylomicron clearance, steroid metabolism, and lipoprotein remodelling ($p_{bon} < 0.05$), and in the obese group, vitamin D metabolism ($p_{bon} = 0.05$), followed by glucuronidation, progesterone response, lipoprotein remodelling, and protein kinase binding ($p_{bon} > 0.1$).

### Discussion

Using a refined environmental UVB measure, we uncovered 162 additional vitamin D status variants in a genome-wide gene-environment

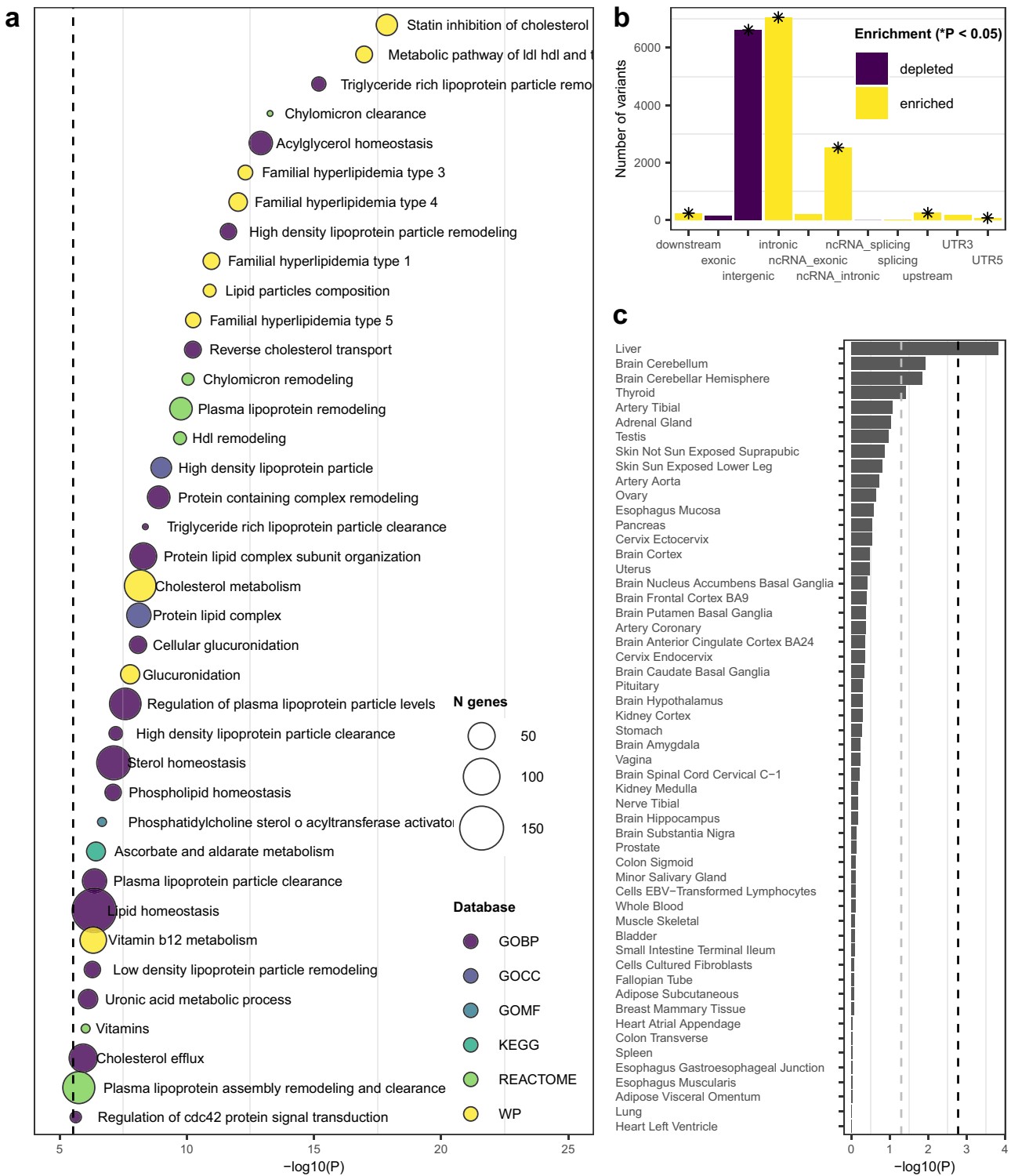

**Fig. 4 | Functional annotation of the 25OHD joint test.** Functional annotation of the joint test results ($p_{joint}$) in FUMA. **a** Gene set analysis in MAGMA, showing significant gene-sets (includes Gene Ontology [GO; BP: biological process, CC: cellular component, MF: molecular function] terms, REACTOME, KEGG, and Wiki-Pathways [WP]). The size of the bubble corresponds to the number of genes in each set and the x-axis shows the −log10(p-value) of the set. The dashed line represents the Bonferroni-corrected p-value (0.05/17010). **b** Enrichment for functional consequence across variants. Enrichment score >1 indicates that the functional consequence is enriched compared to the reference panel (yellow), otherwise it is depleted (purple). An asterisk (*) indicates significant enrichment/depletion (Fisher's p-value < 0.05). **c** Tissue expression analysis of associated variants based on GTEx8 58 tissue types. The grey dashed line represents the one-sided significance threshold at p-value 0.05, and the black line at the corrected p-value (0.05/58; Table S11).

interaction study, doubling the number of known vitamin D loci and replicating the majority of previously reported associations. A third of these loci were low-frequency variants (<5%), many with large effects. Given that several vitamin D status GWA studies have previously been conducted in the same cohort[1–4], the number of newly identified loci is striking. It is well-known that statistical power is a function of effect size, however, less consideration is given to the impact of measurement accuracy and noise. To improve the precision of the

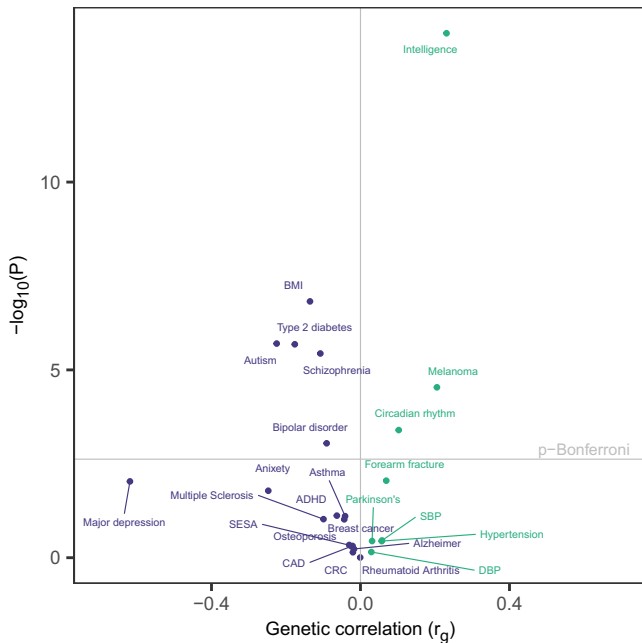

**Fig. 5 | Genetic correlation between 25OHD and selected traits.** Genetic correlation between 25OHD, based on marginal effects, and 25 publicly available GWAS traits. Traits were selected if they have been previously studied in relation to vitamin D status. Genetic correlation ($r_g$) between 25OHD and the phenotype is on the x-axis and the $-\log10(p\text{-value})$ of the association is on the y-axis ($r_g > 0$ shown in green, $r_g < 0$ in purple). The horizontal line represents the Bonferroni-corrected significance threshold (0.05/21) and the vertical line represents $r_g = 0$. CRC colorectal cancer, SBP systolic blood pressure, DBP diastolic blood pressure, CAD coronary artery disease.

environmental exposure in our analysis, in lieu of the commonly used 'season of sample,' we calculated a refined, individualised measurement of ambient UVB. Our findings demonstrate the improvements in the statistical power of GxE studies that can be attained through refined exposure measurement, particularly when coupled with the joint test[12,18].

We report the results of three tests, namely the marginal, interaction, and joint tests. The marginal results are akin to a standard GWAS (G + UVB) and indicate whether there is a significant marginal genetic effect (G). The interaction results refer to the significance of the interaction term (GxE) in a model that includes interaction (G + UVB+GxUVB). Finally, the joint test indicates whether there is a significant main (G) *and/or* interaction (GxUVB) effect, and it was developed as a more powerful alternative where the true model of GxE is not known[12]. Therefore, the results of the joint test provide a more complete picture of the genetic underpinnings of vitamin D status that includes both types of effects, while the interaction test provides direct evidence of gene-environment interactions at a given locus.

Twenty variants, mapping to several known 25OHD genes, were identified using the GxUVB interaction test. Five of these GxE effects were previously reported in candidate gene studies in *SPON1*, *INSC*, *CYP2R1*, *PDE3B*, and *CALCB*[15]. We add novel evidence of interaction in *COPB1* and *PSMA1*, both replicated strongly in the European subgroup and *PSMA1* additionally replicated in LURIC. Interestingly, 16 (80%) of these variants that show significant interaction with UVB are co-located on a 2 MB section of chromosome 11. Future work should explore the mechanism and potential joint mediators of these interactions.

Several of the newly identified genes have highly plausible biological links to the vitamin D pathway, such as *PTH*. Parathyroid hormone (PTH) directly regulates 1-α-hydroxylase, an enzyme that converts 25OHD into 1,25-dihydroxyvitamin D (calcitriol) in the kidney[31]. While

no other 25OHD GWAS reported the *PTH* locus previously, significant associations were identified for other *PTH*-linked variants rs10500783 and rs1459015 in candidate gene studies[32]. *PTH* is also located on the aforementioned region on chromosome 11.

We replicated 9 previously reported UDP-glucuronosyltransferase genes and identified an additional four (*UGT1A1*, *UGT1A3*, *UGT2A1*, *UGT2A2*). The chromosome 2 variant rs35203651 was also nominally significant in the European subgroup replication. *UGT* genes encode enzymes of the glucuronidation pathway in which 25OHD is conjugated into an inactive, more hydrophilic metabolite that can be excreted in urine[33]. We did not replicate a gene involved in a sulphonation pathway (*SULT2A1*)[2]. These findings emphasise the significance of the key elimination pathway. Because back-conversion is possible[2,34,35], these conjugated forms may represent an alternative storage form of vitamin D and thus our findings have implications for how we define vitamin D deficiency, as 25OHD may not be the only 'storage' form of vitamin D.

Using our UVB-adjusted approach, we find that *BMAL1/ARNTL* is associated with vitamin D level, a gene that is also positioned within the aforementioned 2 MB region on chromosome 11. We also replicate the association of *NPAS2*, a *CLOCK* paralogue, and identify an additional variant, rs2282529 in *NPAS4*, a neuronal transcription factor gene that has been implicated in circadian period regulation in a mouse study[36]. Heterodimers containing BMAL1 form the core of the circadian clock, the other partner being either CLOCK or NPAS2 which have overlapping, redundant but not identical, functions[37]. These heterodimers are essential to entrainment of circadian rhythms[38] and our finding of association of multiple elements of the clock provides evidence for an association between circadian rhythm and 25OHD levels.

The effects of perturbation of circadian gene expression have previously been noted[39,40], including in systems associated with vitamin D level such as general metabolism and bone formation[41]. In particular, expression of many lipid metabolism genes have been shown to exhibit circadian fluctuation[42], including cholesterol synthesis. *DHCR7*, which is strongly associated with vitamin D levels as its gene product modulates the level of the 7-dehydrocholesterol precursor, has been shown to be regulated by *BMAL1:NPAS*[43]. Interestingly, the *UGT* genes also show circadian rhythms of expression: for example, some *UGT* genes were down-regulated in Npas2 mouse knockouts[43]. Moreover, previous studies have reported a circadian variation in 25OHD[44-46], and these findings suggest that the glucuronidation pathway may be a factor in how these short-term changes are achieved.

In addition to particular genetic components, previous studies have evaluated heritability in vitamin D status. These have reported a wide range of estimates of the variance explained by genetic effects (from 20 to 90%[24]) and conflicting results of the direction of change of heritability between seasons; for example, Snellman et al.[7] estimated higher heritability in summer, while Karohl et al.[8] estimated higher heritability in winter. We evaluated the relative change in SNP-heritability and observed a clear increase both across UVB quintiles and in a subset of individuals who spent ≥3 h outdoors, supporting an increase in (detectable) genetic effects on 25OHD variance with increasing ambient UVB, whether by lifestyle (time outdoors), or environment (UVB intensity). This is in line with a recent twin study that reported heritability of 0.37 in winter and 0.62 in summer[10].

Our observations of a SNP-heritability gradient and the association between the core circadian clock genes *BMAL1* and *NPAS2* and vitamin D status, is the first evidence to support a mechanistic role for vitamin D in human seasonality. It appears plausible that seasonal variation in circulating vitamin D is not just an index of sunlight exposure but reflects a more complex role in the innate seasonal physiology of humans. Nevertheless, the evidence of a regulatory role for vitamin D in human seasonality is currently purely associative; a causal role will require demonstration of mechanistic involvement.

More broadly, animals at temperate latitudes show vitally important annual fluctuations in metabolic, immune, and reproductive functions in response to predictable seasonal environmental change. The circadian clock contributes to this seasonal flux by tracking day length across the year and regulating the timing of pineal melatonin secretion[47]. Recent genomic and epidemiological studies have provided growing evidence of endogenous seasonal physiology in humans. This was observed in UKB participants across multiple phenotypes, including, among others, metabolism, hormone secretion, immune function, cognition, and BMI, independent of lifestyle and demographic factors, and in line with the endogenous origin suggested by the genetic evidence identified here[48–56]. One study showed seasonal rhythms in mRNA expression, including of *BMAL1*, with corresponding seasonal variation in haematological and inflammatory profiles[48].

The role of vitamin D in seasonal physiology has not been previously considered to our knowledge, but there are numerous potential points for vitamin D signalling in this system. Calcitriol, the active form of vitamin D, can synchronise clock gene expression in vitro[57], and has been shown to increase the secretion of a mediator of seasonality, VGF[58]. With such parallel functions, vitamin D metabolites may play a similar role to a core hormonal regulator of seasonality, melatonin. High melatonin regulates circadian function by signalling night-time[47]. Similarly, low vitamin D status in winter may be a signal of shorter photoperiod that instigates a metabolic, immune, or other phenotype response to darker/colder seasons. If this is the case, seasonal supplementation of vitamin D should be evaluated in terms of a potential impact on innate metabolic rhythms.

In line with previous reports, our results show high overlap between the pathways influencing serum level of vitamin D and many steroid/lipid metabolism processes. Several variants were identified in genes also implicated in lipid/lipoprotein pathways, including genes previously linked with vitamin D status (e.g. *APOE, APOC1, PLA2G3, PCSK9, CELSR2, GALNT2*, and *CETP*)[59] and additional genes, *MOB1B* and *HAVCR1*, which replicated in the European subgroup and in LURIC, respectively[59,60]. The gene-set enrichment analyses additionally prioritised similar pathways and related disorders (e.g. hypercholesterolaemia/hyperlipidemia). The top pathway identified was statin inhibition of cholesterol production, in line with a recent phenome-wide association study, which reported an association between a statin proxy and lower vitamin D levels[61]. When the analysis was stratified by BMI category, the statin-related gene-set was only significant in the overweight category. In contrast, glucuronidation-related gene-sets were in the top results in the normal and obese groups. These results support additional GxE interaction with BMI. The stratified SNP-heritability estimates were very similar across categories, with a slightly higher estimate in the overweight group, suggesting locus-specific interaction(s) rather than genome-wide amplification[62]. We note that bi-directional association likely exists between these traits[63], suggesting that individuals who are vitamin D deficient may also be more susceptible to developing higher BMI or lipid disorders, independent of the effect BMI or lipid metabolism may have on vitamin D status. Seasonality has also been observed in BMI, with higher BMI in winter than in summer, possibly as an essential adaptation for survival in the colder months following sunlight cues[64,65].

Genetic correlation between vitamin D status and other traits was previously examined. For example, Revez et al.[2] identified a significant correlation with several neuropsychiatric phenotypes such as major depression, autism spectrum disorder (ASD), schizophrenia, and others but suggested that these results may be mediated by sunshine exposure. In our UVB-adjusted analysis, we replicated the association with some traits, such as ASD and schizophrenia, but not others, like major depression, suggesting that UVB may in some cases be a confounder of the observed relationships. Observational research supports a link between vitamin D deficiency and a range of diseases related to immunity, cancer, bone health, skin health and others[66,67]. Its potential role in health makes widespread vitamin D deficiency a public

health concern[68]. Although the lack of significant genetic correlation for some outcomes may be a limitation of study design, these results also suggest that vitamin D status may not be genetically linked to these diseases and the observed associations are instead mediated by other exposures, such as UVB or BMI.

The effect sizes we observe are comparable to those reported in previous vitamin D GWAS (Fig. S17). For instance, the newly identified variant in this study that showed the strongest effect was a variant in the *GC* vitamin D-binding protein (rs115366859, $\beta_G = -0.154$ [s.e. 0.010]). The variant with the overall largest effect size in our study was in *PSMA1* (rs577185477, $\beta_G = -0.361$ [0.009]), while the top variant in Manousaki et al.[3] was in *CYP2R1* ($\beta = 0.542$ [0.019]) and in *PDE3B* ($\beta = 0.377$ [0.006]) in Revez et al.[2] We used the White British findings to estimate the genetic score in the European subgroup, to examine clinical utility. On average, the difference in 25OHD concentration between the top and bottom decile was 14.01 nmol/L for the marginal score and 12.07 nmol/L for the interaction score. Newly identified loci can benefit future Mendelian randomisation by improving genetic instrumental variables, particularly to delineate causal relationships between such outcomes, or inform the development of personalised vitamin D dosing approaches.

The present study is primarily based on the data from White British individuals living in the UK, therefore our results are not generalisable to other ethnicities. While we used three replication cohorts, sample sizes were small compared to the discovery cohort and therefore statistical power was limited. Furthermore, replication in the European participants in the UKB cannot be considered independent, because biases related to selection and measurement error are likely correlated to those that may be affecting the discovery sample. The challenge of replication of GWAS results has been discussed at length in the literature[69]. A statistical power calculation[70] based on our discovery cohort effect size and MAF estimated a required median sample size of 103,115 participants to replicate novel variants (*Supplementary Methods*). As such, given the low frequency and small effect size, we did not have enough power to replicate some of the findings presented here, most notably the *PTH* and *BMAL1* variants. Despite the advantage of the more accurate UV exposure measure, the UK's high northerly latitude leads to generally low UVB doses with limited variability. This is even more pronounced in the ORCADES sample from northern Scotland, where both UVB intensity and variance are further reduced, likely leading to decreased power to detect GxE effects. Furthermore, supplement use was not recorded for LURIC or ORCADES participants. Future research should investigate the genetic underpinnings of 25OHD, including GxE interactions, at a wider range and higher UVB radiation levels, both of which could positively affect power—this is supported by the observation of higher SNP-heritability in participants who spend more time outdoors and with increasing ambient UVB dose[71]. Additionally, as the vQTL results in ref. 2 suggest, vitamin D status may be influenced by other environmental exposures. Previous epidemiological studies have reported differences in vitamin D levels by factors such as occupation[72], physical activity[73], or sex[74], which should be evaluated in future GxE studies of this phenotype. Finally, while CW-D-UVB accurately captures the environmental availability of UVB, the actual dose received by each participant will vary depending on clothing, when and how much time they spend outdoors, and other personal and behavioural factors. A more accurate measure of the dose received may be possible with future advances in the available tools, such as scalable dosimeters or linked phone GPS indoor/outdoor data, that would further improve assessment of actual UVB exposure.

In summary, using standardised assessments and a precise ambient UVB dose estimate for each participant, we replicated the majority of previously reported 25OHD SNPs, adding further evidence to support these signals. Novel SNPs remain as putative signals that may be interrogated in future studies through other types of analyses or in independent replication studies when sufficiently large samples

become available. These results expand our knowledge of the genetic aetiology of vitamin D status and demonstrate the advantage of the use of accurate environmental exposure data in genetic studies.

## Methods

### UK Biobank data

The UK Biobank (UKB) is a population-based cohort study of approximately half a million people in the UK. Ethical approval was granted by the North West Multicentre Research Ethics Service Committee[75]. All participants provided informed consent. Participants, aged 37 to 73 years old, were recruited between 2006 and 2010. At their baseline visit, participants completed questionnaires on health and lifestyle, had measurements taken (including height and weight), and provided biological samples. 25-hydroxyvitamin D concentration (25OHD [nmol/L], best biomarker of vitamin D status) was measured using the DiaSorin Chemiluminescent Immunoassay (lower limit of detection was 10 nmol/L) in a blood sample taken at the baseline assessment. To normalise the distribution of raw 25OHD (right-skewed), the variable was log-transformed and standardised to a mean of 0 and standard deviation of 1 (Fig. S2). Individuals with no 25OHD measurement or residential location information were excluded. We adjusted the genetic analysis for 'vitamin D supplement use' (use of vitamin D or multivitamin supplements, yes/no) and for 'fish oil supplement use' (including cod liver oil, yes/no; Fig. S4).

### Ambient UVB

The Tropospheric Emission Monitoring Internet Service (TEMIS; www.temis.nl) database provides daily doses of ultraviolet-B (UVB, kJ/m$^2$). The D-UVB data is determined from a parameterisation of D-UVB as a function of satellite-based ozone observations, the solar zenith angle, and the vitamin D action spectrum (as adopted by the International Commission on Illumination[76]), and includes corrections for surface elevation, surface albedo, sun-earth distance and cloudiness. The method is described and validated by Zempila et al.[77], and has since been upgraded to a higher resolution (see www.temis.nl/uvradiation/product/ for detailed information). Data is provided as averages for each quarter-degree latitude-by-longitude grid cell (at UK latitudes, approximately 28 km [north–south] × 17 km [east–west] each). To facilitate linkage to UVB dose data, participants' home address was used. The Ordinance Survey (OSGB) coordinates used by the UKB for geocoding participants' home location (rounded to 1 km) were first converted to latitude and longitude, using the Transverse Mercator projection functions[78], and then mapped onto the TEMIS grid.

A unique, cumulative and weighted ambient D-UVB value (CW-D-UVB) was calculated for each participant as described previously[6,17] (Fig. S4, *Supplementary Methods*). Briefly, a set of daily corrected ambient D-UVB dose measurements (accounting for cloud cover, ozone layer, and altitude) were retrieved from the TEMIS database, based on each participant's residential location and the date when their blood was collected for vitamin D measurement. To reflect seasonally varying UVB-induced vitamin D production (accumulation when production is abundant as well as ongoing utilisation in physiological processes leading to depletion when vitamin D is scarce), CW-D-UVB is calculated using measured daily D-UVB dose data over a 5-month period up to the date of blood collection (Eq. 1). Daily values are weighted, taking into account the 35-day half-life of vitamin D in the body and so that UVB exposure from a more distant past contributes less to current circulating vitamin D (previous research from our group suggests that UVB contributions to vitamin D status are negligible beyond 135 days prior to sampling)[17].

$$CWDUVB(x) = \sum_{x=1:135} (DUVB(x))^* e^{-\left(\frac{\ln 2}{y}\right)x} \tag{1}$$

Where $x$ is the number of days prior to the date of sampling (from the day before the baseline visit and up to 135 days prior), $y$ is the half-life of the vitamin D-UVB effect in the body at 35 days, and $e^{-(\ln 2/y)x}$ is the weighting formula[17]. The selected parameters are based on previous estimates, which evaluated vitamin D status and UVB in a Scottish cohort (see *Supplementary Methods* and Kelly et al.[17] for more details).

### Quality control

The UKB provides quality-controlled genotype data, which was imputed to the Haplotype Reference Consortium and UK10K haplotype resource panel. Details of the genotyping and imputation methods are available elsewhere[75]. We quality controlled the imputed data in Plink2.0 (Fig. S1)[79]. We first filtered the data for imputation quality (mach-r2-filter 0.8 2.0). In this study, we used the largest ancestry group in the UKB, which is the 'White British' group ($N = 409,522$). The genetic ethnic grouping was performed by the UKB group, and it includes participants who self-identified as 'White British' *and* had similar genetic ancestry in a principal component (PC) analysis. We next filtered for variant and individual missingness >0.02, minor-allele frequency (MAF) > 0.01, and Hardy-Weinberg equilibrium $p$-value > $1 \times 10^{-6}$. Additionally, individuals with missing or mismatched sex, or KING kinship threshold ≥0.0884 (indicating 2nd degree relatives or closer), or heterozygosity ±6 SD were removed.

### Genetic effects analysis

**Association analysis.** We applied a marginal and a genome-wide gene-environment interaction test using GEM1.5[18]. GEM is a tool that implements a generalised linear model (GLM) for large-scale, genome-wide environment interaction studies (GWEIS). Standardised and log-transformed 25OHD was the outcome variable. The marginal model included the following covariates: age, sex, vitamin D supplements, fish oil, CW-D-UVB, and the first 10 principal components (PCs, to account for potential bias from underlying population structure (Eq. 2). The interaction model additionally included a G x CW-D-UVB interaction term. GEM reports marginal, interaction, and joint test results. Results were considered significant at $p$-value < $5 \times 10^{-8}$.

$$25OHD \sim \beta_{marginal}G + age + sex + CWDUVB \\ + supplement + fish\ oil + PC1...PC10 \tag{2}$$

$$25OHD \sim \beta_G G + \beta_{GxE}CWDUVB + age + sex + CWDUVB \\ + supplement + fish\ oil + PC1...PC10 \tag{3}$$

We validated the marginal association results from GEM applying a linear model in Plink2.0 (--glm, no interaction term), similarly adjusting for age, sex, supplements, fish oil, CW-D-UVB dose and PCs; $p < 5 \times 10^{-8}$ was considered significant. Results are comparable across both tools (Fig. S18) so we only report GEM results in the main text.

**Independent variant selection.** *Marginal test:* To identify conditionally independent variants based on the marginal effects, we used the approach in previous vitamin D GWAS[2,3] and applied GCTA-COJO (--cojo-slct) to the results, which uses step-wise selection to select associated SNPs, conditioning on the lead SNP[19]. We used a random sample of 20,000 individuals from the unrelated, White British ancestry population of the UKB, as a linkage disequilibrium (LD) reference sample to account for the linkage structure between variants (10 Mb window). *Interaction and joint test:* To identify independent variants from the interaction and joint test results, we applied the online FUMA pipeline[20]. SNPs were identified as significant at $p$-value < $5 \times 10^{-8}$ and independent at default FUMA $r^2 = 0.1$.

We further compared our results to the available literature (independent variants linked to 25OHD reported in previous vitamin D

GWAS[1–4] and variants in the systematic review on GxE in vitamin D status[15]) and reported novel variants, excluding variants in LD $r^2 > 0.1$.

## Replication

We used three cohorts for the replication analysis: a European subgroup of the UKB, LURIC (hospitalised coronary angiography patients recruited between 1997 and 2000 in Southwestern Germany), and ORCADES (a family-based study that recruited from the Orkney Isles in northern Scotland between 2005 and 2011). Further details of the cohorts are available in the supplementary information. A CW-D-UVB dose was estimated for each participant in the replication cohorts as detailed above. The European UKB subgroup was identified using the results of the Pan-UK Biobank study, which assigned continental ancestry groups based on genetic similarity ($N = 24,235$)[80]. We then filtered out all participants assigned 'White British' ancestry for the replication cohort and quality controlled for imputation quality, sex and genotype missingness, relatedness, and duplicates. Marginal, interaction, and joint tests were similarly applied in GEM for the three replication cohorts, adjusted for age, sex, use of supplements, CW-D-UVB (data on supplement use were not available for LURIC or ORCADES). Tests were further adjusted for 40 PCs to account for the higher population heterogeneity in the European group and for 10 PCs in LURIC. Relatives were excluded in all other cohorts, however, effectively all participants in the ORCADES cohort who were recruited from the isolated Orkney islands were related to some degree. Therefore, standardised log-transformed 25OHD was first regressed against age, sex, genotyping array, the first 10 PCs, and a polygenic random effect to account for relatedness, and the residuals were used as the outcome in the GEM models. We evaluated the agreement in effect size sign between the White British and each of the replication samples.

## Subgroup analyses

**BMI.** BMI was calculated using standard formula and categorised as: normal weight (18.5–24.99 kg/m$^2$), overweight (25–29.99 kg/m$^2$) and obese (≥30 kg/m$^2$). There is evidence of a strong association between 25OHD and BMI (in a model adjusted for age, sex, supplement use and other exposures, $\beta_{BMI} = -0.15$, $p < 0.001$ and $\beta_{CW\text{-}D\text{-}UVB} = 0.35$, $p < 0.001$[81]) and it is possible that 25OHD may influence BMI and vice versa. Additionally, we previously reported an interaction effect between BMI and CW-D-UVB dose on 25OHD[81]. Finally, BMI is also a heritable trait. To avoid collider bias[30], we decided against including BMI as a covariate, similar to previous studies[3]. However, to investigate the role of BMI, we instead performed stratified genetic association analysis by BMI category.

**CW-D-UVB quintiles and 'high time spent outdoors' group.** Previous studies reported varying heritability of 25OHD based on the time of the year[7–10], thus suggesting that genetic contribution to 25OHD level may vary depending on environmental UV intensity. We demonstrated that season only crudely approximates ambient UVB dose (see Fig. S4 and Fig. 3 in ref. 17). Therefore, to investigate this further we stratified our sample into quintiles based on CW-D-UVB, and identified a subgroup of White British individuals who reported spending ≥3 h outdoors in the season of blood sampling ('outdoor group', *Supplementary Methods*).

## SNP-based heritability

We estimated SNP-based heritability ($h^2_{SNP}$) by LD score regression from the marginal results using LDSC[82]. This estimates the proportion of 25OHD variance explained by the genotyped SNPs. We used European LD scores from the 1000 Genomes Project for the LD reference. We also estimated SNP-based heritability by BMI category, CW-D-UVB quintile, and in the outdoor subgroup. Each subgroup was independently analysed with GEM and the marginal effect results were used to estimate $h^2_{SNP}$ in LDSC. We also estimated the variance explained by

the independent COJO SNPs with the formula, variance explained ≈ $2\beta^2 f(1 - f)$, where $\beta$ is the effect estimate and $f$ is the effect allele frequency[83].

## Functional annotation

We performed two primary analyses for functional follow-up, in FUMA and in DAVID, which in turn implement several annotation methods. Results from the joint test from GEM were first annotated with the FUMA online pipeline[20]. The FUMA application output includes gene-based, gene-set, and tissue-based annotations (MAGMA). The gene-based analyses annotate genes associated with 25OHD on a genome-wide level in addition to enrichment statistics of the functional consequences of the input variants (e.g. UTR, exonic, downstream). MAGMA gene-set and tissue-based enrichment analyses identify pathways or tissues enriched in the 25OHD-associated variants. To specifically identify potential functional consequences of the interaction variants, we also applied the FUMA pipeline to the results from the interaction test. Similarly, to identify potential novel pathways associated with 25OHD, we performed gene-based annotation based only on the list of the significant independent variants identified from the marginal, interaction, and/or joint tests (in contrast, the FUMA annotation is based on the full summary statistics from the joint test). These variants were annotated with the Ensembl Variant Effect Predictor (VEP)[25]. The output list of genes from VEP was further annotated with DAVID[27]. From the DAVID output, we report KEGG pathway and clustered GO term annotations (biological processes, molecular functions, and cellular components) and clustered enrichment of DISGENET disease terms. For all DAVID annotations, we used an enrichment threshold EASE 0.05. GO terms and disease terms are clustered to reduce redundant annotations (default parameters including similarity threshold 0.50, initial group membership 3, and multiple linkage threshold 0.50).

## Genetic correlation with other outcomes

We evaluated the genetic correlation between 25OHD and selected health and disease outcomes based on the marginal summary statistics using bivariate LD score regression, applied in LDSC (we chose marginal effects as only one effect estimate value could be included)[82]. We included traits that we were interested in evaluating from the traits reported in the recent 25OHD GWA[2] and phenome-wide Mendelian randomisation[28] studies of vitamin D, with open access summary statistics available. GWAS summary statistics were downloaded primarily from the GWAS Catalog, the Psychiatric Genetics Consortium, and the Complex Trait Genetics Lab ('Data availability'). Genetic correlations were considered significant at a Bonferroni-corrected threshold of 0.002.

## Reporting summary

Further information on research design is available in the Nature Portfolio Reporting Summary linked to this article.

# Data availability

Summary statistics for the marginal, interaction, joint, and stratified (BMI, CW-D-UVB quintiles, time outdoors) analyses are available from the GWAS Catalog (GCST90652548:GCST90652559). This research has been conducted using the UK Biobank Resource under Application Number 73479. The CW-D-UVB dose (application ID 73479) and participant data are available by application to the UK Biobank. UV data is available at www.temis.nl/uvradiation. GWAS summary statistics for genetic correlation can be downloaded from the GWAS Catalog (study accession) https://www.ebi.ac.uk/gwas/ for multiple sclerosis (GCST005531), melanoma skin cancer (GCST90011809), circadian rhythm (GCST003837), asthma (GCST010042), osteoporosis (GCST90018887), Parkinson's disease (GCST009325), colorectal cancer (CRC; GCST9012950, Alzheimer's disease (GCST90027158), intelligence (GCST004364), breast cancer (GCST007236), BMI (GCST90179150), DBP (GCST90000059), forearm

fracture (GCST90281273), and SBP (GCST90000062); from the Psychiatric Genetics Consortium https://pgc.unc.edu/for-researchers/download-results/ for autism spectrum disorder (asd2019), schizophrenia (scz2022), bipolar disorder (bip2021), major depression disorder (mdd2018), and anxiety (anx2016); from https://plaza.umin.ac.jp/yokada/datasource/software.htm for rheumatoid arthritis; from https://cncr.nl/research/summary_statistics/ for sensitivity to environmental stress and adversity (SESA); from the Program in Complex Trait Genomics https://cnsgenomics.com/content/data for type 2 diabetes; from http://www.cardiogramplusc4d.org/data-downloads/ for coronary artery disease (CARDIoGRAM); and from Surendran et al., 2020 https://doi.org/10.1038/s41588-020-00713-x for hypertension.

## Code availability

R code for CW-D-UVB calculation is available at https://github.com/rshraim/UVdose.

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

## Acknowledgements

We would like to thank Professor Michael Gill for his valuable feedback on the manuscript. Calculations were performed on the Tinney cluster maintained by the Trinity Centre for High-Performance Computing (Research IT). This cluster is funded by a Grant from Research Ireland to Professor Romero-Ortuno under Grant number 18/FRL/6188. Genotyping in LURIC was supported by the 7th Framework Programmes Atheroremo (Grant Agreement number 201668) and RiskyCAD (grant agreement number 305739) of the European Union. The Orkney Complex Disease Study (ORCADES) was supported by the Chief Scientist Office of the Scottish Government (CZB/4/276, CZB/4/710), a Royal Society URF to J.F.W., the MRC Human Genetics Unit quinquennial programme 'QTL in Health and Disease', Arthritis Research UK and the European Union framework programme 6 EUROSPAN project (contract no. LSHG-CT-2006-018947). DNA extractions were performed at the Edinburgh Clinical Research Facility, University of Edinburgh. We would like to acknowledge the invaluable contributions of the research nurses in Orkney and the administrative team in Edinburgh. Finally, we thank the participants of the UKB, LURIC, and ORCADES cohorts. R.S. is supported by Research Ireland through the Research Ireland Centre for Research Training in Genomics Data Science under grant number 18/CRT/6214 and in part EU's Horizon 2020 research and innovation programme under the Marie Sklodowska-Curie grant H2020- MSCA-COFUND-2019-945385; M.D. by the CRUK Programme grant - DRCPGM\100012; C.W. and L.M.L. by the European Research Council (ERC) under the European Union's Horizon 2020 research and innovation programme (grant agreement No 950010); E.T. by CRUK Career Development Fellowship (C31250/A22804); and L.M.L. by the European Research Council (ERC) under the European Union's Horizon 2020 research and innovation programme (grant agreement No 950010) and with the financial support of Taighde Éireann – Research Ireland, under Grant number 21/RC/10294_P2 at FutureNeuro Research Ireland Centre for Translational Brain Science.

## Author contributions

R.S. and L.Z. conceived the study and designed the analyses. R.S. conducted the analyses. L.Z., R.M., M.D., E.T., M.T., J.v.G and M.v.W. provided advice on the methodology. C.W. and L.M.L. provided advice on the interpretation of the results. R.R.O. provided computational resources. J.v.G and M.v.W provided TEMIS data, and M.E.K., S.P., W.M., B.S.F., and J.F.W. supplied the LURIC and ORCADES cohorts and supported the analysis based on these samples. R.M. and L.Z. supervised the project. R.S., R.M., and L.Z. wrote the manuscript with participation from all authors. All authors reviewed and approved the final paper.

## Competing interests

The authors declare no competing interests.

## Additional information

[1]Department of Public Health and Primary Care, Institute of Population Health, Trinity College Dublin, Dublin, Ireland. [2]Department of Clinical Medicine, Trinity Translational Medicine Institute, Trinity College Dublin, Dublin, Ireland. [3]The Research Ireland Centre for Research Training in Genomics Data Sciences, University of Galway, Galway, Ireland. [4]Medical Research Council Human Genetics Unit, Institute of Genetics and Molecular Medicine, University of Edinburgh, Edinburgh, UK. [5]Epidemiology, Biostatistics and Biodemography, Department of Public Health, University of Southern Denmark, Odense, Denmark. [6]Danish Institute for Advanced Study, University of Southern Denmark, Odense, Denmark. [7]Department of Biology, Maynooth University, Kildare, Ireland. [8]The Kathleen Lonsdale Human Health Institute, Maynooth University, Kildare, Ireland. [9]Royal Netherlands Meteorological Institute, 3731 GA De Bilt, The Netherlands. [10]Discipline of Medical Gerontology, School of Medicine, Trinity College Dublin, Dublin, Ireland. [11]Mercer's Institute for Successful Ageing, St James's Hospital, Dublin, Ireland. [12]Global Brain Health Institute, Trinity College Dublin, Dublin, Ireland. [13]FutureNeuro Research Ireland Centre, Department of Biology, Maynooth University, Maynooth Co., Kildare, Ireland. [14]SYNLAB MVZ für Humangenetik, Mannheim, Germany. [15]Department of Internal Medicine, Division of Endocrinology and Diabetology, Medical University of Graz, Graz, Austria. [16]SYNLAB Academy, SYNLAB Holding Deutschland GmbH, Mannheim and Augsburg, Germany. [17]Medical Clinic III (Cardiology, Angiology, Pneumology), University of Heidelberg, Heidelberg, Germany. [18]Centre for Global Health Research, Usher Institute, University of Edinburgh, Edinburgh, Scotland, UK. [19]Centre for Genomic and Experimental Medicine, Institute of Genetics and Cancer, University of Edinburgh, Edinburgh, Scotland, UK. [20]Cancer Research UK Edinburgh Centre, Institute of Genetics and Cancer, University of Edinburgh, Edinburgh, UK. [21]Centre for Global Health, Usher Institute, University of Edinburgh, Edinburgh, UK. ✉e-mail: zgagal@tcd.ie

