## [Transparent Peer Review file · Nature Communications]

Genome-wide gene-environment interaction study uncovers 162 vitamin D status variants using a precise ambient UVB measurement

Corresponding Author: Professor Lina Zgaga

Version 0:

Reviewer comments:

Reviewer #1

(Remarks to the Author)

The author used UKB data ($n=338,977$) to conduct a GWAS on 25OHD concentrations, incorporating a test for interaction with UVB dose. This dose was calculated as a cumulative score of UVB radiation exposure over 135 days before the sampling date, based on participants' residential location, with data extracted from the Tropospheric Emission Monitoring Internet Service database. The analysis included the marginal, main, interaction, and joint effects, identifying 307 independent genetic variants, of which 162 were novel. This nearly doubles the number of 25(OH)D associated variants from previous GWAS on vitamin D. Besides replicating previously identified SNPs, the study uncovered additional variants and identified loci biologically linked to 25(OH)D. A notable example is PTH, a locus for parathyroid hormone, which regulates 1-alpha-hydroxylase, the enzyme that converts 25(OH)D to its active and functionally relevant form, 1,25-dihydroxyvitamin D. The gene set analysis identified biologically relevant processes, including links with lipid metabolic pathways. Overall, this is an interesting study with the novel idea of using composite score of UVB radiation exposure as part of the interaction analyses, which resulted multiple novel variants of biological relevance.

Major

1. Methods. The construction of the cumulative and weighted UVB dose was not clearly presented. Given that this is a core part of the study, clear and detailed information on how this exposure was constructed is crucial. According to the author, the UVB dose was constructed using the residential address and date of sampling for each participant. It appears that the date of sampling wasn't included in the score but was used to gather the prior 5 months of daily UV radiation doses based on their residential locations. If this is true, the statement about how the date of sampling was used in constructing the UVB dose should be made explicit. Furthermore, the weighting strategy and the source of 35-day half-life of vitamin D data, as well as the logic behind its selection, are not clear from the description. Also, link is given for the UVB data source, but it is unclear if the source code and variable created will be made available to use by others as per the UKB policy?

2. Although some of the identified loci have plausible biological links, the lack of independent replication is a clear limitation. Efforts to replicate the identified signals in separate cohort(s) would add notable value to the study. For marginal effects this could be easily done in the GWAS from the SUNLIGHT consortium, which also should provide a source for relevant independent replication cohorts for the interaction analyses.

3. Adjustment strategy: There are discrepancies in the statements for model adjustments, and while the method section indicates adjustments for 10 PCs, in several places elsewhere in the text these are not mentioned. Please check throughout and ensure all statements are systematic and correct.

Typical UKB adjustment to account for population structure includes 40 PCs + birth location (and includes assessment centre, see Haworth et al. PMID: 30659178). Can the authors confirm that this difference in the adjustment strategy does not influence their findings? We appreciate that the UVB index is likely to correlate with assessment centre and birth location, however, it is unclear why such a limited set of PCs was used.

4. The author reported 22 novel variants, including PTH, from the marginal effect model. It is surprising that these variants

weren't identified in the prior Revez et al. GWAS, given the use of a similar cohort and nearly similar modelling. Can the author comment on this? Could the author fit an exactly similar model, including the adjustment strategy used by Revez et al., to see if the associations are lost? Does the relatively relaxed approach to control for population structure (as noted above), contribute to the differences in the number of associated variants?

Minor

1. Terminology. Please define vitamin D status. Please check throughout that vitamin D related terminology is accurate and used appropriately for the context. Eg relevant for the abstract, 'vitamin D' per se is inactive while the hormonal form (will have various functions 1,25OH₂D).

2. The introduction does not acknowledge previous efforts in genomewide GxE analyses on 25OHD levels. For instance, Revez et al. [PMID: 32242144] used the season of blood draw to test for interaction, identifying six independent variants; some related analyses also done by the SUNLIGHT consortium. Were the previously identified interacting variants replicated in your interaction model

3 The gene set analysis identified biologically relevant processes, including lipid metabolic pathways and Statin drug targets. The relationship of the latter with 25OHD was also identified in prior PheWAS of lipid-lowering drug targets using the UKB (Pham et al. PMID: 37208559), which should be referenced in the paper.

4. Figure 1. Please explicitly state what kind of stratification analyses were performed. Stratified GWAs?

5. Figure 3, label is not clear. Please be explicit with what is meant by "based on the marginal effect estimates and p-values for each"?

6. Figure 5 can be simplified by using two colours for reflecting the positively versus negatively correlated traits.

7. For clarity of presentation, it would be better to modify Fig S1 into panels. Panel A could show the flow of participants, while Panel B could show the flow of included SNPs in the analysis, indicating how many SNPs were excluded due to each QC criterion.

8. Lines 122-125: Can the author comment on why they choose to report IQR for the effect estimates while the estimates in Table S1 is as beta and SE?

9. Reference 26 seems to be incorrect, as it is a bidirectional study on lipid levels, while the bidirectional association with BMI has been published elsewhere (Vimalaswaran et al. PMID: 23393431).

Reviewer #2

(Remarks to the Author)

This manuscript investigates the genetic basis underlying vitamin D status using data from the UK Biobank. A novel approach is applied to identify genetic variants associated with vitamin D levels, which includes a gene-environment interaction term to account for UVB radiation exposure. This leads to the identification of 307 independent genetic variants, of which 162 are novel. These are followed up and annotated using functional analysis.

The novel variants confirm the relevance of lipid metabolism and highlight the regulation of the circadian rhythm as a potential new pathway.

Further, it is shown that the genetic heritability is exposure-dependent, with higher heritability in those exposed to higher UVB radiation levels.

The role of gene-environment interactions in complex phenotypes and the difficulty of detecting them has been discussed previously. The approach proposed here is therefore a valuable contribution introducing a new perspective to the ongoing debate and offers many possibilities for future analyses.

I have a few suggestions and comments, mainly to improve clarity.

Results

1. Figure 1: for each step, the sample size should be included

2. GEM results: the supplement includes a Venn diagram showing the overlap between the COJO, interaction and joint tests. The numbers cannot be directly replicated with the numbers given in the main text. It would be helpful for the reader if the Venn diagram could be shown in the main paper as well.

3. Page 4: "Novel genes identified through significant marginal effects..." It is unclear which method was used for gene annotation. This paragraph might be moved to "functional annotation"

4. The formula used for the calculation of the SNP-based heritability ($2\beta^2 f(1-f)$) should be described in the main text including the annotations of the abbreviations for β and f

5. The effect size and units are not clear. For example, for the genetic risk score calculation described on page 5 the beta estimates are given but it is not clear whether the score was standardised or how it is distributed. Also, for the log-normal dependent variable, it might be more useful to give an interpretable transformation such as percent change

6. The selection of the traits included for calculation of the genetic correlation is not clear and should be described in more details in the methods section.

7. BMI: I agree with the authors' concerns about the collider bias and the approach that was subsequently applied to the BMI analysis. However, the paragraph is rather short and difficult to follow due to the large amount of numbers. Some of the supplementary figures are also not correctly referenced.

Discussion

1. The discussion is very well written and brings up many interesting aspects to provide new angles for future research.
2. The technical aspects of the chosen GEM model and relevance of main, interaction and joint effects for the interpretation could be more elaborated. This is only briefly mentioned in the results (direction of interaction effects)
3. The implications of the BMI analysis (including heritability) should be further considered in the discussion. Especially whether some differences were found here in relation to the lipid metabolism (page 9, second paragraph)
4. The two paragraphs on genetic correlation and the link with other diseases related to vitamin D deficiency (page 9) should be combined. Many of these diseases (e.g. asthma, bone health) were included in the genetic correlation analysis but did not show a significant association. This seems contradictory. How would the authors interpret this observation?
5. The only reference to the effect size is found as a limitation of generally low UVB radiation. It would be interesting if the authors could include 1-2 sentences on the effect size and potential clinical implications

Methods

1. UVB radiation: It would be helpful for the reader if the authors could describe in 1-2 sentences how the data was generated and how the models were derived by TEMIS. For instance, were the models validated, and if yes, how? Is there a unit available for UVB radiation?
2. Genetic effect analysis
 - Did the authors further test the role of sex (e.g. are there sex specific differences in the heritability?). Are there other covariates available which have been described in the context of vitamin D status, e.g. physical activity?
 - The model used here with the three components marginal, interaction and joint effects is not intuitively clear. It might be easier to follow if the formula (Equation 1) could be shown for the models used (i.e. Eq1 for the marginal model, Eq 2 for the joint/interaction model) and then highlight the terms for which the beta coefficients are reported.

Supplementary material

1. Not all figures and tables included in the supplement are also cited in the main text. Additionally, the supplement is not structured by figures and tables nor by the order they appear in the text. It would be easier for the reader to use the resources provided when this could be re-structured to follow the main text.
2. Figure S1: The boxes seem blurry. It would be useful to add number of SNPs and/or individuals to each step
3. Figure S7: it is stated that the GEM and Plink results are "largely comparable". How was this determined and are there variants that differ?
4. Tables are numbered starting with 1 both in the pdf as well as excel files. This should be corrected.
5. Table S1 (excel spreadsheet "S5_Sig.Ind._SNPs") says:
 - Genes could not be differently color-coded within individual cells so where an asterisk is present in a highlighted cells only genes marked with an asterisk (*) are novel: CRCT1, MROH2A, UGT1A3, UGT1A1, MORF4L1P3, C11orf58.

The annotation with asterisk is sufficient (but it would be possible to colour code only part of a cell's text). Therefore, I suggest removing that part and shorten the footnote to:

- Genes marked with an asterisk (*) are novel: CRCT1, MROH2A, UGT1A3, UGT1A1, MORF4L1P3, C11orf58.

Reviewer #3

(Remarks to the Author)

The authors conducted a genome-wide association study (GWAS) of 25-hydroxyvitamin D concentration (25OHD) using a refined solar radiation measure to account for the influence of ambient solar ultraviolet B radiation (UVB) exposure and explored the gene-environment (GxE) interaction in depth. Overall, the study found an elevated SNP-based heritability of 25OHD, multiple novel loci both marginal and interactional and relevant biological pathways of the newly identified genes. These findings are of interest, but there are concerns that need clarification.

1. The individual representation of UVB exposure is thoughtful and novel. However, the UVB exposure is quite random depending largely on one's outdoor time and sun protections. Did the individualized representation of one's actual exposure to UVB outperform the averaged measurement of seasonal exposure? This needs detailed clarification.
2. Would it make more sense if a conventional epidemiological association study (for example, a regression model) was conducted prior to the GWAS? Moreover, a regression model helps to distinguish if the interaction is additive or multiplicative.
3. In the genetic correlation section, what is the novelty using the marginal effects? What does this add as previous studies have done on this topic?

Reviewer #4

(Remarks to the Author)

I co-reviewed this manuscript with one of the reviewers who provided the listed reports. This is part of the Nature Communications initiative to facilitate training in peer review and to provide appropriate recognition for Early Career

Researchers who co-review manuscripts.

Version 1:

Reviewer comments:

Reviewer #1

(Remarks to the Author)
Replication.

In the discussion the authors note "The present study uses data from individuals of European descent, therefore our results are not generalisable to other ethnicities." However, in their interpretation on what represents 'a signal', they do not appear to have considered replication at all, and looking at the results, about half of the signals do not appear to be replicating/generalisable within the European ancestry populations included in the UK biobank.

The summary statements included in the discussion about 'replication' are not helpful for assessing the relevance of the individual signals. Looking at Table S6 and using a statistically very lenient(!) approach where any of the three tests (marginal, joint or interaction) being significant at $p < 0.05$ would represent 'replication', for nearly half of the SNP (~150) there was no statistical support. An argument that a sample of ~24,000 would be insufficient in size for this type of lenient replication analysis is not easy to digest, and this does suggest that many of the 'signals' may be spurious. What would be interesting, however, is to see which of the variants are replicated and where the evidence appears to be robust. This is where I would have liked to see the paper being focused on and this is where the possible value of this paper would lie.

While I appreciate their effort to replicate (and their note on the difficulties to achieve independent replication in this particular context), it is important to acknowledge that while replication in the European sub-sample of the UK biobank will ensure that a single individual is not included both in the discovery and replication, this does not make it 'independent' replication. Biases related to selection and measurement error are likely to be directly correlated between the sample non-British Europeans and white British participants in the UK Biobank, which the least should be mentioned as a limitation.

Supplementary table S6. There is a typo as the authors note that "Significant p-values $< 5 \times 10^{-8}$ are shown in bold.", however, many of the bolded p-values are larger than this.

Some adjustments to interpretation which have been included in the limitations, have not been incorporated into the title or interpretation/summary (end of discussion). Despite acknowledging deficits of their area based UVB measures as measures of actual dosage received by the individual, they claim to have incorporated data on a "precise ambient UVB dose" and "accurate exposure data". Please revise.

Reviewer #2

(Remarks to the Author)
All the comments have been adequately addressed. I do not have further suggestions.

Reviewer #3

(Remarks to the Author)

Reviewer #4

(Remarks to the Author)
I co-reviewed this manuscript with one of the reviewers who provided the listed reports. This is part of the Nature Communications initiative to facilitate training in peer review and to provide appropriate recognition for Early Career Researchers who co-review manuscripts.

Version 2:

Reviewer comments:

Reviewer #1

(Remarks to the Author)
No further comments.

Reviewer #4

(Remarks to the Author)
I co-reviewed this manuscript with one of the reviewers who provided the listed reports. This is part of the Nature

Communications initiative to facilitate training in peer review and to provide appropriate recognition for Early Career Researchers who co-review manuscripts.

RESPONSE TO REVIEWER COMMENTS

[*Blame it on the Sunshine: Genome-wide gene-environment interaction study uncovers 162 novel vitamin D status variants after precise assessment of ambient UVB exposure.*]

Note to reviewers: Thank you for your time and insightful feedback. Point-by-point responses are provided below and, where applicable, the updated text is copied into the response for convenience.

Reviewer #1:	1
Reviewer #2:	8
Reviewer #3:	14
Reviewer #4:	17
References	17

Reviewer #1:

The author used UKB data (n=338,977) to conduct a GWAS on 25OHD concentrations, incorporating a test for interaction with UVB dose. This dose was calculated as a cumulative score of UVB radiation exposure over 135 days before the sampling date, based on participants' residential location, with data extracted from the Tropospheric Emission Monitoring Internet Service database. The analysis included the marginal, main, interaction, and joint effects, identifying 307 independent genetic variants, of which 162 were novel. This nearly doubles the number of 25(OH)D associated variants from previous GWAS on vitamin D. Besides replicating previously identified SNPs, the study uncovered additional variants and identified loci biologically linked to 25(OH)D. A notable example is PTH, a locus for parathyroid hormone, which regulates 1-alpha-hydroxylase, the enzyme that converts 25(OH)D to its active and functionally relevant form, 1,25-dihydroxyvitamin D. The gene set analysis identified biologically relevant processes, including links with lipid metabolic pathways. Overall, this is an interesting study with the novel idea of using composite score of UVB radiation exposure as part of the interaction analyses, which resulted multiple novel variants of biological relevance.

We thank the Reviewer for their time required to complete this review and for their insightful feedback and comments. We have taken these onboard and believe it has improved our manuscript.

Major

1. Methods. The construction of the cumulative and weighted UVB dose was not clearly presented. Given that this is a core part of the study, clear and detailed information on how this exposure was constructed is crucial. According to the author, the UVB dose was constructed using the residential address and date of sampling for each participant. It appears that the date of sampling wasn't included in the score but was used to gather the prior 5 months of daily UV radiation doses based on their residential locations. If this is true, the statement about how the date of sampling was used in constructing the UVB dose should be made explicit.

We have expanded our description of the calculation of the UVB dose in the **Methods** section ('Ambient UVB exposure') and additional details have been provided in the **Supplementary Methods**. We also clarify that the UVB dose is indeed based on the 5 months prior to the date

of sampling and does not include the day of sampling itself. The modified text in the main document reads as follows (see also Reviewer 2:Comment 13):

“A unique, cumulative and weighted ambient D-UVB value (CW-D-UVB) was calculated for each participant as described previously^{6,17} (Fig. S4). Briefly, a set of daily ambient D-UVB dose measurements (restricted to wavelengths that can induce vitamin D synthesis in the skin and accounting for cloud cover, ozone layer, altitude and other) were retrieved from the TEMIS database, based on each participant’s residential location and the date when their blood was collected for vitamin D measurement. To reflect seasonally varying UVB-induced vitamin D production (accumulation when production is abundant as well as ongoing utilisation in physiological processes leading to depletion when vitamin D is scarce), CW-D-UVB is calculated using measured daily D-UVB dose data over a 5-month period up to the date of blood collection (Eq1). Daily values are weighted, assuming a 35-day half-life so that UVB exposure from a more distant past contributes less to current vitamin D status (previous research from our group suggests that UVB contributions to vitamin D status are negligible beyond 135 days prior to sampling).¹⁷

$$CWDUVB(x) = \sum_{x=1:135} (DUVB(x)) * e^{-\left(\frac{\ln 2}{y}\right)x}$$

Equation 1

*Where x is the number of days prior to the date of sampling (from the day before the baseline visit and up to 135 days prior), y is the half-life of the vitamin D UVB effect in the body (35 days), and $e^{-\left(\frac{\ln 2}{y}\right)x}$ is the weighting formula.¹⁷ The parameters are based on previous estimates which evaluated vitamin D status and UVB in a Scottish cohort (see **Supplementary Methods** and Kelly et al¹⁷ for more details).”*

Furthermore, the weighting strategy and the source of 35-day half-life of vitamin D data, as well as the logic behind its selection, are not clear from the description.

The weighting function is needed to account for vitamin D accumulation in the body when UVB intensity is high, and depletion during the winter when solar radiation is too weak for vitamin D synthesis to occur. Consequently, we typically observe peak daily UVB exposure in June/July while 25OHD concentration peaks 1-2 months later, in August/September in the Northern hemisphere. Thus, we don’t expect a strong correlation between UVB dose on the day of blood draw and 25OHD.

Biological data suggests that the half-life of vitamin D in the body is about 2 months, while circulating 25OHD has been reported to be broken down after 15 days (Jones, 2008). These observations were the starting point for earlier work which sought to determine an UVB-based estimate that would most strongly correlate to 25OHD concentration (Kelly et al., 2016). Several scenarios were examined, including periods of 30-240 days before date of blood draw and half-lives of 21 to 49 days, with final parameters selected based on performance and complexity. These estimates were used in several other cohorts and found to be very strongly correlated to 25OHD in each (e.g. (Brennan et al., 2024; O’Sullivan et al., 2019; Scott et al., 2022)).

Also, link is given for the UVB data source, but it is unclear if the source code and variable created will be made available to use by others as per the UKB policy?

The CW-D-UVB dose data that we calculated for each participant will be uploaded to the UK Biobank on publication of this work, as per UK Biobank policy, and available as per UKB access procedure. The source code will be made available upon request. Furthermore, we are collaborating with TEMIS to present the code as an accessible R package. We expect to release this in due course, which will facilitate calculations of cumulative and weighted UV variables based on TEMIS and also other UV data sources.

2. Although some of the identified loci have plausible biological links, the lack of independent replication is a clear limitation. Efforts to replicate the identified signals in separate cohort(s) would add notable value to the study. For marginal effects this could be easily done in the GWAS from the SUNLIGHT consortium, which also should provide a source for relevant independent replication cohorts for the interaction analyses.

We agree with the Reviewer that a replication would further strengthen the findings. For our replication cohort, we used the *European* sample from the UK Biobank, comprising N=24,235 participants. This cohort includes White, non-British participants (our main analysis was restricted to White *British* participants). We considered the benefits of the larger sample size (SUNLIGHT N=79,000), however several significant concerns drove our decision to use the European sample from the UKB instead. SUNLIGHT is a consortium that included cohorts from multiple countries (i.e. different lifestyle, level of ambient UV, genetic background), with studies performed at different times (i.e. population likelihood to take vitamin D supplement), using different data collection methods (e.g. 25OHD assay, diet/supplements questionnaires). This would notably increase the heterogeneity in the replication cohort. Furthermore, for variables that are not available for all cohorts, we would either need to simplify the model (e.g. remove the vitamin D supplement covariate), or exclude those cohorts from the analysis, both of which would erode the benefits of the larger sample size SUNLIGHT may offer. A major specific concern relates to the UVB data. To calculate CW-D-UVB, we would need a residential address for participants, which may not be available or sharable. Furthermore, TEMIS only covers Europe, which means that TEMIS could not be used for e.g. Northern American cohorts included in SUNLIGHT.

Therefore, to replicate the main results in the European UKB sample, we applied the marginal, interaction, and joint tests implemented in GEM to the 307 significant independent variants that were originally identified in the discovery cohort (i.e. White British group). Two of the variants were not available in the European data as they did not pass QC, therefore, we compare the results of 305 variants. We use the same model but include the first 40 PCs because of the greater genetic heterogeneity of the European UKB sample. The approach to replication we use is similar to the Revez et al. 2020 study (referenced in Comment 4 below): we compare the direction of effect and correlation of effect estimates. We find that our results replicate strongly. **Table S6** has been added with details of the results for the 305 SNPs and **Fig. S6** shows the correlation between effect estimates.

Methods: “We used a European subgroup of the UKB (N=24,235) for replication analysis. The larger European group was identified using the results of the Pan-UK Biobank study, which assigned continental ancestry groups based on genetic similarity.⁷³ We then filtered out

all participants assigned 'White British' ancestry for the replication cohort and quality controlled for imputation quality, sex and genotype missingness, relatedness, and duplicates. A CW-D-UVB dose was estimated for each participant as detailed above. Marginal, interaction, and joint tests were similarly applied in GEM, adjusted for age, sex, use of supplements, CW-D-UVB, and 40 PCs to account for the higher population heterogeneity in this sample. We evaluated the agreement in effect size sign between the White British and European samples."

Results: *"We used the European cohort of the UKB (N=24,235, mean age: 55.7 y, 54.2% female) for replication. 305 of the 307 genome-wide significant variants remained in this cohort after quality control. We compared the direction of the effect estimate in each (p_{binom} represents the p-value from a binomial test with random sign of effect estimates as the null hypothesis; **Fig. S6**). The same sign was observed for 95 of the 104 marginal variants (91%, $p_{\text{binom}} = 2.99 \times 10^{-19}$), 17 of the 20 interaction variants (85%, $p_{\text{binom}} = 0.0026$) and 226 of the 236 joint variants (comparing the direction of the main effect estimate β_G [Eq3], 96%, $p_{\text{binom}} = 2.31 \times 10^{-54}$). The correlation between the White British and European effect estimates was highly significant in each of the marginal (correlation = 0.98), interaction (0.93), and joint (0.93) tests."*

3. Adjustment strategy: There are discrepancies in the statements for model adjustments, and while the method section indicates adjustments for 10 PCs, in several places elsewhere in the text these are not mentioned. Please check throughout and ensure all statements are systematic and correct.

Thank you for bringing this to our attention. The revised draft has been updated to clarify the adjustment for 10 PCs, and to include this information where relevant.

Typical UKB adjustment to account for population structure includes 40 PCs + birth location (and includes assessment centre, see Haworth et al. PMID: 30659178). Can the authors confirm that this difference in the adjustment strategy does not influence their findings? We appreciate that the UVB index is likely to correlate with assessment centre and birth location, however, it is unclear why such a limited set of PCs was used.

We adjusted for **10 PCs** given the characteristics of our study sample. Our discovery cohort included White British only, a single ancestry group we expect to be relatively homogeneous - adjusting for 10 PCs is generally considered sufficient for more genetically homogeneous populations and is commonly used, e.g. in other UKB studies (Alcalde-Herraiz et al., 2024; Lane et al., 2016; Yang et al., 2024) and in the GEM methods paper (Westerman et al., 2021). As the reviewer notes, the UVB dose correlates with the **assessment centre**, since it is based on the geographical location of the participants. This is evidenced by statistically significant association: in a simple regression model, CW-D-UVB ~ assessment_center (ref: Croydon), almost all assessment centres ($p < 2 \times 10^{-6}$) were significantly associated with the CW-D-UVB dose (except Leeds and Swansea). Additionally, 25OHD is a stable analyte, study procedures were consistent across centres, and vitamin D samples were all processed in the same facility. Finally, for **birth location**, we consider two main reasons that might warrant adjustment for this variable: either to reduce confounding due to population structure, or to account for 'geographically heterogeneous complex traits' as described in the Haworth et al paper. For vitamin D status, a complex trait which is largely determined by recent periods of a single environmental exposure, we believe such confounding through population structure is

adequately adjusted for by the PCs while the UVB dose captures key environmental variation relevant for this trait.

Nonetheless, we acknowledge that in larger studies small genetic or other differences may have a measurable impact on association results, in which case adjusting for more PCs/covariates may improve results and, most importantly, potentially reduce false positives. Thus, we performed a sensitivity analysis and compared the significant independent results from our original models (adjusted for UVB, supplements, age, sex, and 10 PCs) to model 2 (further adjusted for 40 PCs) and model 3 (further adjusted for 40 PCs, birth location, and assessment centre).

Number of significant independent hits:

Test	Original	Model 2	Model 3
GxE	20 SNPs	20 SNPs 100% replicated	18 SNPs 90% replicated (17 identical, 1 in LD)
joint	238 SNPs	239 SNPs 96% replicated (221 identical, 8 in LD)	241 SNPs 92% replicated (203 identical, 16 in LD)
marginal	105 SNPs	103 SNPs 96% replicated (98 identical, 3 in LD)	104 SNPs 92% replicated (88 identical, 9 in LD)

The results are largely unchanged. Of the SNPs that did not replicate, many had not mapped to any gene in the original results (e.g. 8 of the joint SNPs) and key variants like those that mapped to *PTH*, *NPAS*, and *BMAL1* variants appeared as significant independent across models. Therefore, the conclusions based on the original model hold and to avoid introducing collinear variables (e.g. UVB dose, assessment centre), we kept the original analysis. The results reported in this comment have been added to the **Supplementary Material**.

4. The author reported 22 novel variants, including *PTH*, from the marginal effect model. It is surprising that these variants weren't identified in the prior Revez et al. GWAS, given the use of a similar cohort and nearly similar modelling. Can the author comment on this? Could the author fit an exactly similar model, including the adjustment strategy used by Revez et al., to see if the associations are lost? Does the relatively relaxed approach to control for population structure (as noted above), contribute to the differences in the number of associated variants?

Naturally, any change to the model, QC process, participant or variable selection will influence the results. For example, results from the two most recent UKB vitamin D GWAS studies (Manousaki et al., 2020; Revez et al., 2020) report notably different findings. When we look at results from the marginal model (with methods comparable to Revez et al., except that we included UVB variable), the number of significant independent variants we find is in fact smaller compared to the Revez et al study. Since the Revez study used the COJO method to select independent variants from their marginal model, we applied this to our marginal results also, for comparability. We report 105 COJO variants from the marginal model, while Revez et al report 143. Based on these findings and the results from the additional analysis (Comment 3 above), we can safely conclude that adjusting for 10 PCs did not lead to an inflation of positive findings. Most [novel] hits were identified in the joint test, which is sensitive to the improved E measure. Looking at the **Fig. 2b/c** of the revised manuscript, one can compare the "yield" from the marginal (pink) and joint (blue) models: 50 independent

variants were found in both marginal and joint test (49+1, as not looking at interaction test here [yellow]), 55 in marginal only, and additional 188 in joint test only.

Many of these novel variants point to genes such as *PTH*, which have definitive biological links with vitamin D that were previously demonstrated through experimental studies, although surprisingly not identified through genome-wide scans. Therefore, we interpret the difference in results to be primarily due to the investigation of interactions (most notably through joint test) and the use of the ambient UVB dose, an improved environmental exposure. Compared to season or month of assessment, this continuous outcome is more granular and less confounded by differences in UVB arising from location, or time of year.

Minor

1. Terminology. Please define vitamin D status. Please check throughout that vitamin D related terminology is accurate and used appropriately for the context. Eg relevant for the abstract, ‘vitamin D’ per se is inactive while the hormonal form (will have various functions 1,25OH₂D).

Thank you for this point. Vitamin D terminology has been checked throughout. Most notably, we can confirm that all instances where ‘vitamin D’ was used to mean ‘25-hydroxyvitamin D’ have been amended. In some instances we use ‘vitamin D status’ to refer to the measured 25OHD concentration in the blood (this has been stated in the text).

2. The introduction does not acknowledge previous efforts in genomewide GxE analyses on 25OHD levels. For instance, Revez et al. [PMID: 32242144] used the season of blood draw to test for interaction, identifying six independent variants; some related analyses also done by the SUNLIGHT consortium.

Were the previously identified interacting variants replicated in your interaction model

The introduction has been updated to clearly acknowledge previous genome-wide efforts and evidence of GxE in vitamin D research, with an explicit mention of the Revez et al interaction analysis. The expanded text now reads as follows:

“Earlier GWAS and candidate gene vitamin D studies reported evidence of GxE and noted the significant challenges of integrating environmental data into genetic studies.¹⁵ For instance, Revez et al 2020 conducted a genome-wide variance quantitative trait locus (vQTL) analysis and identified 25 independent variants, only 5 of which showed evidence of interaction with season (others were reported as candidates for GxE with other exposures), while Manousaki et al 2020 evaluated interaction with season only in the GWAS significant variants. Environmental exposures relevant for vitamin D status are variable and exposures over longer periods of time matter. Yet, the prevailing approach is to only use the season of blood draw as a proxy of environmental exposure. This is a rather imprecise, transient approximation of solar radiation, which varies substantially by day of the year, latitude, altitude, ozone, cloud cover and other factors. For example, in London on average a 50-fold difference was observed between annual high and annual low (average daily UVB dose was 0.11 kJ/m² in December and 5.56 kJ/m² in June), which illustrates the large variability.¹⁶”

In terms of GxE interaction replication, we use the recent GxE systematic review (Shraim et al 2022) as a point of comparison to help us identify novel interactions, and replicate GxE

interaction in 5 previously reported genes, *SPON1*, *INSC*, *CYP2R1*, *PDE3B*, and *CALCB*. The following text has been added to the **Discussion**:

“Twenty variants, mapping to several known 25OHD genes, were identified using the GxUVB interaction test. Five of these GxE effects were previously reported in candidate gene studies: in SPON1, INSC, CYP2R1, PDE3B, and CALCB. We add novel evidence of interaction in COPB1 and PSMA1.¹⁵”

3. The gene set analysis identified biologically relevant processes, including lipid metabolic pathways and Statin drug targets. The relationship of the latter with 25OHD was also identified in prior PheWAS of lipid-lowering drug targets using the UKB (Pham et al. PMID: 37208559), which should be referenced in the paper.

Thank you for pointing out this relevant work. The **Discussion** section on the gene-set analysis has been updated to include the PheWAS paper.

“The top pathway identified was statin inhibition of cholesterol production, in line with a recent phenome-wide association study, which reported an association between a statin proxy and lower vitamin D levels.⁵⁵”

4. Figure 1. Please explicitly state what kind of stratification analyses were performed. Stratified GWAs?

The figure caption has been updated:

“We performed genome-wide association and interaction analyses in the ‘White British’ group, and then conducted stratified genome-wide association and interaction analyses separately in several subgroups: by BMI category (normal, overweight and obese), by quantile of ambient UVB exposure, and in participants who spend 3 hr or more outdoors per day.”

5. Figure 3, label is not clear. Please be explicit with what is meant by “based on the marginal effect estimates and p-values for each”?

SNP-heritability was estimated using the results of the marginal test. The figure label has been updated to clarify this:

“Figure 3: SNP-based heritability [h^2_{SNP} (standard error)] was calculated based on the results of the marginal test for the entire cohort and within each subgroup”

6. Figure 5 can be simplified by using two colours for reflecting the positively versus negatively correlated traits.

Thank you for the suggestion. The figure has been updated as suggested and is clearer now.

7. For clarity of presentation, it would be better to modify Fig S1 into panels. Panel A could show the flow of participants, while Panel B could show the flow of included SNPs in the analysis, indicating how many SNPs were excluded due to each QC criterion.

Thanks for this. The figure has been updated as suggested.

8. Lines 122-125: Can the author comment on why they choose to report IQR for the effect estimates while the estimates in Table S1 is as beta and SE?

We originally reported IQR to show the spread of the effect estimates, but we appreciate this is inconsistent with our other summary statistics so we have updated this to mean and SD instead.

9. Reference 26 seems to be incorrect, as it is a bidirectional study on lipid levels, while the bidirectional association with BMI has been published elsewhere (Vimaleswaran et al. PMID: 23393431).

Thanks for spotting this, the reference has been corrected.

Reviewer #2:

This manuscript investigates the genetic basis underlying vitamin D status using data from the UK Biobank. A novel approach is applied to identify genetic variants associated with vitamin D levels, which includes a gene-environment interaction term to account for UVB radiation exposure. This leads to the identification of 307 independent genetic variants, of which 162 are novel. These are followed up and annotated using functional analysis.

The novel variants confirm the relevance of lipid metabolism and highlight the regulation of the circadian rhythm as a potential new pathway. Further, it is shown that the genetic heritability is exposure-dependent, with higher heritability in those exposed to higher UVB radiation levels. The role of gene-environment interactions in complex phenotypes and the difficulty of detecting them has been discussed previously. The approach proposed here is therefore a valuable contribution introducing a new perspective to the ongoing debate and offers many possibilities for future analyses. I have a few suggestions and comments, mainly to improve clarity.

We would like to thank the Reviewer for their time reviewing our manuscript. The suggestions and comments provided were helpful and have helped us improve the clarity and quality of our work.

Results

1. Figure 1: for each step, the sample size should be included

Figure 1 has been amended to include this information.

2. GEM results: the supplement includes a Venn diagram showing the overlap between the COJO, interaction and joint tests. The numbers cannot be directly replicated with the numbers given in the main text. It would be helpful for the reader if the Venn diagram could be shown in the main paper as well.

The authors agree, the Venn diagram has been moved to the Main text as an additional panel in **Figure 2**. The text reports: “At a genome-wide significance level of $p < 5 \times 10^{-8}$, we identified 953 variants with interaction effects, 11,509 variants with marginal effects, and 11,715 variants with joint effects (**Fig. 2,S5**)” and in the Venn diagram these are broken down to show the overlap between different tests (e.g. $953 = 5 + 171 + 777$, and so on). Hopefully this is clearer now. Whilst addressing this comment we also realised that two of the variants had two alternative alleles and were mistakenly counted twice in the total number of significant variants. This has been corrected throughout.

3. Page 4: “Novel genes identified through significant marginal effects...” It is unclear which method was used for gene annotation. This paragraph might be moved to “functional annotation”

A line has been added to clarify that gene annotation was performed in Ensembl’s Variant Effect Predictor (VEP) and the paragraph was moved to the ‘functional annotation’ section as suggested.

4. The formula used for the calculation of the SNP-based heritability ($2\beta^2 f(1-f)$) should be described in the main text including the annotations of the abbreviations for β and f

Thank you for spotting this. β and f have now been defined in the main text.

5. The effect size and units are not clear. For example, for the genetic risk score calculation described on page 5 the beta estimates are given but it is not clear whether the score was standardised or how it is distributed. Also, for the log-normal dependent variable, it might be more useful to give an interpretable transformation such as percent change

We agree with the reviewer that the effect size results are difficult to interpret for the general reader because of the variable transformations. This includes effect sizes of the (i) individual SNPs (based on log-transformed and standardised 25OHD) and (ii) the genetic score. For SNP effect size, we have added a column in **Table S5** showing the percent change in 25OHD based on the marginal effect estimates. In relation to genetic risk score, the effect estimates from the regression models are now reported in the original 25OHD unit (nmol/L). We previously reported ‘average of genetic score’ in the supplementary data, we now report ‘sum of score’ and include included a figure of the overall distribution of genetic scores in the European cohort as well as the change in 25OHD by decile of genetic score in the supplementary file (**Fig. S12**). The description of the genetic risk score been updated as follows.

*“Genetic risk scores were calculated based on the marginal effects (β_{marginal}) and on the main and interaction effects ($\beta_{G+CW-D-UVB} \cdot \beta_{G \times E}$) estimated in the White British group. In an age and sex adjusted model, both scores were significantly associated with 25OHD in the European group ($p < 10^{-100}$, **Table S4**). 25OHD increased by 14.319 nmol/L (s.e. 0.485) for every unit increase in the marginal score (mean 0.43; s.d. 0.28) and by 5.21 nmol/L (s.e. 0.214) for the interaction score (mean 0.262; s.d. 0.647; **Fig. S12** shows the distribution of the genetic scores).”*

We have also added a comparison to SNP effect estimates from the two recent vitamin D GWAS.

“The effect sizes we observe are comparable to those reported in previous vitamin D GWAS (Fig. S17). For instance, the novel variant that showed the strongest effect was a variant in the GC vitamin D-binding protein (rs115366859, $\beta_G = -0.154$ [s.e. 0.010]). The variant with the overall largest effect size in our study was in PSMA1 (rs577185477, $\beta_G = -0.361$ [0.009]), while the top variant in Manousaki³ was in CYP2R1 ($\beta = 0.542$ [0.019]) and in PDE3B ($\beta = 0.377$ [0.006]) in Revez.²”

6. The selection of the traits included for calculation of the genetic correlation is not clear and should be described in more details in the methods section.

This has been further clarified in the **Methods** section. We selected traits that we are interested in evaluating, including some traits that were tested in recent 25OHD GWA (Revez et al., 2020) and phenome-wide Mendelian randomisation (Meng et al., 2019) studies of vitamin D, with available open access summary statistics.

“We included traits that we were interested in evaluating from the traits reported in recent 25OHD GWA² and phenome-wide Mendelian randomisation²⁵ studies of vitamin D, with open access summary statistics available.”

7. BMI: I agree with the authors' concerns about the collider bias and the approach that was subsequently applied to the BMI analysis. However, the paragraph is rather short and difficult to follow due to the large amount of numbers. Some of the supplementary figures are also not correctly referenced.

It is reassuring that the Reviewer agrees with us on the issue of collider bias and the approach to handling BMI. The paragraph has been simplified. Instead of providing the number of significant SNPs for each BMI category and test (marginal, joint, interaction) in the text, we show the full details of these numbers in the **Supplementary Material (Fig. S15, S16)**. In the main text we provide percentages for the overlap in the joint and the interaction variants so that it is easier to follow and compare, which is also better aligned to the key point we are making (that notable differences suggest presence of an interaction with the BMI).

Discussion

8. The discussion is very well written and brings up many interesting aspects to provide new angles for future research.

Thank you for your positive feedback on the discussion. We appreciate that you found it interesting and see how it opens new directions for future research.

9. The technical aspects of the chosen GEM model and relevance of main, interaction and joint effects for the interpretation could be more elaborated. This is only briefly mentioned in the results (direction of interaction effects)

The authors agree that this would be a helpful addition to the discussion. We have added a paragraph to the **Discussion** section elaborating on the differences between the marginal, interactions, and joint tests and utility of each test.

“We report the results of three tests, namely the marginal, interaction and joint tests. The marginal results are akin to a standard GWAS (G+UVB) and indicate whether there is a significant marginal genetic effect (G). The interaction results refer to the significance of the interaction term (GxE) in a model that includes interaction (G+UVB+GxUVB). Finally, the joint test indicates whether there is a significant main (G) and/or interaction (GxUVB) effect, and it was developed as a more powerful alternative where the true model of GxE is not known.¹² Therefore, the results of the joint test provide a more complete picture of the genetic underpinnings of vitamin D status that includes both types of effects, while the interaction test provides direct evidence of gene-environment interactions at a given locus.”

10. The implications of the BMI analysis (including heritability) should be further considered in the discussion. Especially whether some differences were found here in relation to the lipid metabolism (page 9, second paragraph)

We believe the reviewer is referring to the paragraph beginning “In line with previous reports...” (in our version, the second paragraph on page 9 discusses circadian rhythm). We have added a discussion of the top gene-sets and the SNP-heritability estimates from the BMI-stratified analysis.

“The top pathway identified was statin inhibition of cholesterol production, in line with a recent phenome-wide association study, which reported an association between a statin proxy and lower vitamin D levels.⁵⁵ When the analysis was stratified by BMI category, the statin-related gene-set was only significant in the overweight category. In contrast, glucuronidation-related gene-sets were in the top results in the normal and obese groups. These results support additional GxE interaction with BMI. The stratified SNP-heritability estimates were very similar across categories, with a slightly higher estimate in the overweight group, suggesting locus-specific interaction(s) rather than genome-wide amplification.⁵⁶”

11. The two paragraphs on genetic correlation and the link with other diseases related to vitamin D deficiency (page 9) should be combined. Many of these diseases (e.g. asthma, bone health) were included in the genetic correlation analysis but did not show a significant association. This seems contradictory. How would the authors interpret this observation?

We have combined the paragraphs as suggested. While the lack of a significant result cannot be interpreted as a lack of genetic correlation between such outcomes, it may also be that previously observed links with other diseases have been mediated by confounders that were not properly adjusted for in the models, or that the observed link between vitamin D deficiency and these diseases is not due to genetic causes. We have added text to **Discussion** as follows:

“Genetic correlation between vitamin D status and other traits was previously examined. For example, Revez et al., 2020 identified a significant correlation with several neuropsychiatric phenotypes such as major depression, autism spectrum disorder (ASD), schizophrenia, and others but suggested that these results may be mediated by sunshine exposure. In our UVB-adjusted analysis, we replicated the association with some traits, such as ASD and schizophrenia, but not others, like major depression, suggesting that UVB may in some cases be a confounder of the observed relationships. Observational research supports a link between vitamin D deficiency and a range of diseases related to immunity, cancer, bone

health, skin health and others.^{60,61} Its potential role in health makes widespread vitamin D deficiency a public health concern.⁶² Although the lack of significant genetic correlation for some outcomes may be a limitation of study design, these results also suggest that vitamin D status may not be genetically linked to these diseases and the observed associations are instead mediated by other exposures, such as UVB or BMI.”

12. The only reference to the effect size is found as a limitation of generally low UVB radiation. It would be interesting if the authors could include 1-2 sentences on the effect size and potential clinical implications

It seems that this comment is referring to the second last paragraph in the **Discussion** section.

“Despite the advantage of the more accurate UV exposure measure, the high northerly latitudes of the UK mean that the distribution of UVB doses is generally low, and variability limited. Future research should investigate the genetic underpinnings of 25OHD, including GxE interactions, at a wider range and higher radiation levels, both of which could positively affect power.⁶⁰”

We have added text to refer to clinical utility and comment on the effect sizes. Please also see response to Reviewer 2:Comment 5.

“The effect sizes we observe are comparable to those reported in previous vitamin D GWAS (Fig. S17). For instance, the novel variant that showed the strongest effect was a variant in the GC vitamin D-binding protein (rs115366859, $\beta_G = -0.154$ [s.e. 0.010]). The variant with the overall largest effect size in this study was in PSMA1 (rs577185477, $\beta_G = -0.361$ [0.009]), while the top variant in Manousak et al.³ was in CYP2R1 ($\beta = 0.542$ [0.019]) and in PDE3B ($\beta = 0.377$ [0.006]) in Revez et al.² We used the White British findings to estimate the genetic score in the European subgroup, to examine clinical utility. On average, the difference in 25OHD concentration between the top and bottom decile was 14.01 nmol/L for the marginal score and 12.07 nmol/L for the interaction score. Newly identified loci can benefit future Mendelian randomisation by improving genetic instrumental variables, particularly to delineate causal relationships between such outcomes, or inform the development of personalised vitamin D dosing approaches.”

Methods

13. UVB radiation: It would be helpful for the reader if the authors could describe in 1-2 sentences how the data was generated and how the models were derived by TEMIS. For instance, were the models validated, and if yes, how? Is there a unit available for UVB radiation?

The authors agree. UVB is measured in kJ/m^2 , this and further details of the TEMIS data have been added to the **Methods** text and the following text to the **Supplementary Materials** (see also Reviewer 1:Comment 1).

“**TEMIS data.** The D-UVB data is determined from a parameterisation of D-UVB as a function of satellite-based ozone observations, the solar zenith angle, and the vitamin D action spectrum (as adopted by the International Commission on Illumination²), and includes

corrections for surface elevation, surface albedo, sun-earth distance and cloudiness. The method is described and validated by Zempila et al.³; this has since been upgraded to a higher resolution (see www.temis.nl/uvradiation/product/ for detailed information). The date of blood draw was extracted for each participant. Based on residential location and sample date, an array of daily D-UVB doses was extracted from the TEMIS database. Unlike most other biomarkers, accumulation of vitamin D in the body can be observed in the summer - i.e. times of the high UVB intensity, and diminution in the winter when solar radiation is too weak for vitamin D synthesis to occur. Previous work determined that an UVB-based estimate that most strongly correlates to 25OHD concentration uses data covering a period of 135 days prior to blood sampling, whilst weighting the exposures so that more recent exposures contribute more to the estimate.⁴ To weigh UVB dose, we use the recommended half-life of 35 days (which is in accordance with previously reported observations: the half-life of vitamin D in the body has been reported to be about 2 months, while circulating 25OHD has been reported to be broken down after 15 days⁴). Thus, cumulative and weighted UVB (CW-D-UVB) was calculated as $CW-D-UVB(x) = \sum_{x=1:135}(D-UVB(x) * e^{-(\ln 2/y)x})$ (Eq1, described in the main **Methods**)."

14. Genetic effect analysis

- Did the authors further test the role of sex (e.g. are there sex specific differences in the heritability?). Are there other covariates available which have been described in the context of vitamin D status, e.g. physical activity?

Thank you for this. Our research question, with focus on environmental sunshine, gave rise to the UVB-stratified and 'outdoor groups' analyses, and was motivated by the contradictory literature on varying vitamin D heritability by season. Similarly, the complex and likely bidirectional relationship between vitamin D and BMI motivated the BMI-stratified analysis. We agree with the reviewer and think it would be interesting to perform further stratified analysis such as by sex and physical activity, and other factors like smoking or chronic illness, however this is unfortunately beyond the scope of this paper. We have included a note on other exposures to the **Discussion**:

"Additionally, as the vQTL results in Revez et. al. 2020 suggest, vitamin D status may be influenced by other environmental exposures. Previous epidemiological studies have reported differences in vitamin D levels by factors such as occupation,⁶⁴ physical activity,⁶⁵ or sex,⁶⁶ which should be evaluated in future GxE studies of this phenotype."

- The model used here with the three components marginal, interaction and joint effects is not intuitively clear. It might be easier to follow if the formula (Equation 1) could be shown for the models used (i.e. Eq1 for the marginal model, Eq 2 for the joint/interaction model) and then highlight the terms for which the beta coefficients are reported.

The **Methods** section has been updated as suggested. Discussion that further clarifies this has been provided. See response to Reviewer 2:Comment 9.

Supplementary material

15. Not all figures and tables included in the supplement are also cited in the main text. Additionally, the supplement is not structured by figures and tables nor by the order they appear in the text. It would

be easier for the reader to use the resources provided when this could be re-structured to follow the main text.

The supplementary file has been restructured to start with methods and then the results section is divided into tables and figures, with the list of SNPs, FUMA, and LDSC tables remaining in the Excel sheet document. The references to supplementary figures and tables in the main text have been updated to match the new order.

16. Figure S1: The boxes seem blurry. It would be useful to add number of SNPs and/or individuals to each step

Thanks, the figure has been updated. See also Reviewer1:Minor Comment 7.

17. Figure S7: it is stated that the GEM and Plink results are “largely comparable”. How was this determined and are there variants that differ?

We compared the variants identified in each analysis, only one variant was different. The detailed description of these results has been added to the figure S7 caption.

18. Tables are numbered starting with 1 both in the pdf as well as excel files. This should be corrected.

Thank you, this has now been corrected.

19. Table S1 (excel spreadsheet “S5_Sig.Ind._SNPs”) says:

- Genes could not be differently color-coded within individual cells so where an asterisk is present in a highlighted cells only genes marked with an asterisk (*) are novel: CRCT1, MROH2A, UGT1A3, UGT1A1, MORF4L1P3, C11orf58.

The annotation with asterisk is sufficient (but it would be possible to colour code only part of a cell’s text). Therefore, I suggest removing that part and shorten the footnote to:

- Genes marked with an asterisk (*) are novel: CRCT1, MROH2A, UGT1A3, UGT1A1, MORF4L1P3, C11orf58.

We originally colour coded only part of the cell as the reviewer suggests, however, unfortunately, depending on the software used to open the file, the cell was sometimes automatically recoloured. So, to avoid potential information loss during file transfer, we annotated the cells with the asterisk. We have shortened the footnote as follows: “*In cells with novel and known genes, those marked with an asterisk (*) are novel: CRCT1, MROH2A, UGT1A3, UGT1A1, MORF4L1P3, C11orf58.*”

Reviewer #3:

The authors conducted a genome-wide association study (GWAS) of 25-hydroxyvitamin D concentration (25OHD) using a refined solar radiation measure to account for the influence of ambient solar ultraviolet B radiation (UVB) exposure and explored the gene-environment (GxE) interaction in depth. Overall, the study found an elevated SNP-based heritability of 25OHD, multiple

novel loci both marginal and interactional and relevant biological pathways of the newly identified genes. These findings are of interest, but there are concerns that need clarification.

We would like to thank the Reviewer for taking the time to review our manuscript. The feedback and insights are highly valuable and have helped us improve the quality and clarity of our work.

1. The individual representation of UVB exposure is thoughtful and novel. However, the UVB exposure is quite random depending largely on one's outdoor time and sun protections. Did the individualized representation of one's actual exposure to UVB outperform the averaged measurement of seasonal exposure? This needs detailed clarification.

We agree with the reviewer that personal factors affect how much of the environmentally available UVB the participant actually receives. We agree that the time outdoors variable is important, which is why we conducted stratified analysis amongst those who reported spending >3h/day outdoors. It is likely that having the actual "dose received" would further improve models, however we don't believe that a notable improvement beyond an accurate estimate of the ambient dose presented here is possible with the current data or tools.

Two key components are relevant for UVB-induced synthesis of vitamin D: (1) the intensity of UVB that is available in the environment (ambient UVB dose) and (2) individual downstream factors that affect the utilisation of this UVB and determine the "dose received". These factors are strongly dependent on individual characteristics and behaviours, such as time of day spent outdoors (midday hours matter most), clothing choices, sunscreen application, tan, activity level (movement may stimulate skin circulation), etc. It is difficult to capture all of these changing factors accurately, particularly over longer periods of time. The average hours spent outdoors per day in winter vs. summer among UKB participants underscores notable seasonal differences (145,303 reported spending ≥ 3 hr/day outdoors in summer compared to 47,931 in winter).

While neither "season of measurement" nor the "ambient UVB dose" fully capture the differences resulting from personal factors outlined above, the latter is a significantly more accurate measure of the environmental availability of UVB (e.g. the 'season' variable equates summer in Glasgow and summer in London). The adjusted R^2 from a linear regression model adjusted for the UVB dose ($R^2=0.2197$) was higher than that of a similar model adjusted for time spent outdoors in the season of measurement ($R^2=0.1014$) or for season of measurement ($R^2=0.1142$; standardised $\log(25\text{OHD}) \sim \text{age} + \text{sex} + \text{supplement} + \text{fish oil} + [\text{season or UVB or time outdoors}]$). We previously examined the relationship of many of those personal and behavioural factors with circulating 25OHD concentration and observed the strongest association with CW-D-UVB (Brennan et al., 2024), which we attribute to this variable being largely unconfounded and unbiased. Thus, we believe that the uniform approach to assess ambient UVB exposure for each participant outweighs the benefits of adding other noisy and/or biased variables. In the future, it may be possible to use GPS data to track actual duration and time-of-day spent outdoors over longer periods of time, which would benefit analyses like these. The following section was added to the **Discussion**.

"Finally, while CW-D-UVB accurately captures the environmental availability of sunshine, the actual dose received by each participant is likely to vary depending on clothing, when and

how much time they spend outdoors, and other personal and behavioural factors. A more accurate measure of the dose received may be possible with future advances in the available tools, such as scalable dosimeters or linked phone GPS data, that would further improve assessment of UVB exposure.”

2. Would it make more sense if a conventional epidemiological association study (for example, a regression model) was conducted prior to the GWAS? Moreover, a regression model helps to distinguish if the interaction is additive or multiplicative.

We have undertaken these analyses and published this epidemiological study recently (Brennan et al., 2024). We evaluated UVB dose, time spent outdoors, and BMI among other predictors in a linear regression model as well as interaction between UVB dose and other factors. There was evidence of several complex interactions influencing vitamin D level. In general, the conventional approach to the study of GxE in vitamin D has focused on candidate genetic variants comparing effect estimates across genotypes rather than a genome-wide test. Previous research has been published in this area by other groups (e.g. (Engelman et al., 2013; Robien et al., 2013)).

In terms of distinguishing between additive or multiplicative interaction: we wanted to perform an exploratory genome-wide test without restriction to preselected variants, comparable to other GWAS (e.g. Revez et al 2020). The GEM model tests for additive interaction with the product term in the linear regression for a continuous outcome such as 25OHD (Westerman et al., 2021; Westerman & Sofer, 2024). In general, interaction can be evaluated on different scales and the interpretation differs accordingly, although this is most relevant for binary outcomes and this is an ongoing point of discussion in the field (VanderWeele & Knol, 2014).

3. In the genetic correlation section, what is the novelty using the marginal effects? What does this add as previous studies have done on this topic?

The marginal model is equivalent to a standard GWAS that does not include an interaction term. In the marginal effects model we present, the key difference to previous vitamin D GWAS is that the model was additionally adjusted for the UVB dose, which can improve the genetic analysis. Our previous epidemiological analysis identified CW-D-UVB to be most strongly associated with 25OHD among White participants in the UKB (Brennan et al., 2024). Statistically, this means that CW-D-UVB explains a large proportion of variance in the model, which helps to uncover the effects of weaker genetic predictors, because less variance remains unexplained. Without this adjustment, the effect of the genetic variants may be obscured by the UVB exposure. Therefore, controlling for UVB in the model enables a more precise evaluation of the contribution of the genetic background to the outcome of interest, i.e. 25OHD concentration. This approach ensures that the genetic association is not misestimated due to unaccounted variation.

Genetic correlation aims to estimate the overlap in genetic background between different heritable outcomes, such as 25OHD concentration and other disease outcomes. Previous studies found a genetic correlation between 25OHD and other traits, e.g. depression, but it was unclear whether this was due to confounding by sunshine. The novelty here is that by adjusting for sunshine our analysis in part reduces this confounding. For example, in the most

recent vitamin D GWAS (Revez et al 2020) when evaluating genetic correlation between 25OHD and cognitive-related phenotypes, the authors noted that their “*findings may be mediated by an association between higher intelligence and behaviour associated with less exposure to bright sunshine (and thus, lower 25OHD).*” The significant correlation with depression was lost in our analysis, supporting the explanation that the association of 25OHD with some traits may indeed be mediated by sunshine (please also see response to Reviewer 2:Comment 11).

Reviewer #4:

We would like to express our appreciation to this Reviewer for their time and attention spent co-reviewing our manuscript and helping to enhance the quality of the work.

References

- Alcalde-Herreraiz, M., Català, M., Prats-Urbe, A., Paredes, R., Xie, J., & Prieto-Alhambra, D. (2024). Genome-wide association studies of COVID-19 vaccine seroconversion and breakthrough outcomes in UK Biobank. *Nature Communications*, 15(1), 8739. <https://doi.org/10.1038/s41467-024-52890-6>
- Brennan, M. M., van Geffen, J., van Weele, M., Zgaga, L., & Shraim, R. (2024). Ambient ultraviolet-B radiation, supplements and other factors interact to impact vitamin D status differently depending on ethnicity: A cross-sectional study. *Clin Nutr*, 43(6), 1308-1317. <https://doi.org/10.1016/j.clnu.2024.04.006>
- Engelman, C. D., Meyers, K. J., Iyengar, S. K., Liu, Z., Karki, C. K., Igo, R. P., Jr., Truitt, B., Robinson, J., Sarto, G. E., Wallace, R., Blodi, B. A., Klein, M. L., Tinker, L., LeBlanc, E. S., Jackson, R. D., Song, Y., Manson, J. E., Mares, J. A., & Millen, A. E. (2013). Vitamin D intake and season modify the effects of the GC and CYP2R1 genes on 25-hydroxyvitamin D concentrations. *Journal of Nutrition*, 143(1), 17-26. <https://doi.org/10.3945/jn.112.169482>
- Jones, G. (2008). Pharmacokinetics of vitamin D toxicity. *Am J Clin Nutr*, 88(2), 582S-586S. <https://doi.org/10.1093/ajcn/88.2.582S>
- Kelly, D., Theodoratou, E., Farrington, S. M., Fraser, R., Campbell, H., Dunlop, M. G., & Zgaga, L. (2016). The contributions of adjusted ambient ultraviolet B radiation at place of residence and other determinants to serum 25-hydroxyvitamin D concentrations. *Br J Dermatol*, 174(5), 1068-1078. <https://doi.org/10.1111/bjd.14296>
- Lane, J. M., Vlasac, I., Anderson, S. G., Kyle, S. D., Dixon, W. G., Bechtold, D. A., Gill, S., Little, M. A., Luik, A., Loudon, A., Emsley, R., Scheer, F. A. J. L., Lawlor, D. A., Redline, S., Ray, D. W., Rutter, M. K., & Saxena, R. (2016). Genome-wide association analysis identifies novel loci for chronotype in 100,420 individuals from the UK Biobank. *Nature Communications*, 7(1), 10889. <https://doi.org/10.1038/ncomms10889>
- Manousaki, D., Mitchell, R., Dudding, T., Haworth, S., Harroud, A., Forgetta, V., Shah, R. L., Luan, J., Langenberg, C., Timpson, N. J., & Richards, J. B. (2020). Genome-wide Association Study for Vitamin D Levels Reveals 69 Independent Loci [Research Support, Non-U.S. Gov't]. *American Journal of Human Genetics*, 106(3), 327-337. <https://doi.org/https://dx.doi.org/10.1016/j.ajhg.2020.01.017>
- Meng, X., Li, X., Timofeeva, M. N., He, Y., Spiliopoulou, A., Wei, W. Q., Gifford, A., Wu, H., Varley, T., Joshi, P., Denny, J. C., Farrington, S. M., Zgaga, L., Dunlop, M. G., McKeigue, P., Campbell, H., & Theodoratou, E. (2019). Phenome-wide Mendelian-randomization study of genetically determined vitamin D on multiple health outcomes using the UK Biobank study. *Int J Epidemiol*, 48(5), 1425-1434. <https://doi.org/10.1093/ije/dyz182>

- O'Sullivan, F., Raftery, T., van Weele, M., van Geffen, J., McNamara, D., O'Morain, C., Mahmud, N., Kelly, D., Healy, M., O'Sullivan, M., & Zgaga, L. (2019). Sunshine is an Important Determinant of Vitamin D Status Even Among High-dose Supplement Users: Secondary Analysis of a Randomized Controlled Trial in Crohn's Disease Patients. *Photochemistry and Photobiology*, 95(4), 1060-1067. <https://doi.org/https://doi.org/10.1111/php.13086>
- Revez, J. A., Lin, T., Qiao, Z., Xue, A., Holtz, Y., Zhu, Z., Zeng, J., Wang, H., Sidorenko, J., Kemper, K. E., Vinkhuyzen, A. A. E., Frater, J., Eyles, D., Burne, T. H. J., Mitchell, B., Martin, N. G., Zhu, G., Visscher, P. M., Yang, J., . . . McGrath, J. J. (2020). Genome-wide association study identifies 143 loci associated with 25 hydroxyvitamin D concentration [Article]. *Nature Communications*, 11(1). <https://doi.org/10.1038/s41467-020-15421-7>
- Robien, K., Butler, L. M., Wang, R., Beckman, K. B., Walek, D., Koh, W. P., & Yuan, J. M. (2013). Genetic and environmental predictors of serum 25-hydroxyvitamin D concentrations among middle-aged and elderly Chinese in Singapore [Observational Study]. Research Support, Non-U.S. Gov't]. *British Journal of Nutrition*, 109(3), 493-502. <https://dx.doi.org/10.1017/S0007114512001675>
- Scott, J., Havyarimana, E., Navarro-Gallinad, A., White, A., Wyse, J., van Geffen, J., van Weele, M., Buettner, A., Wanigasekera, T., Walsh, C., Aslett, L., Kelleher, J. D., Power, J., Ng, J., O'Sullivan, D., Hederman, L., Basu, N., Little, M. A., Zgaga, L., . . . groups, U. (2022). The association between ambient UVB dose and ANCA-associated vasculitis relapse and onset. *Arthritis Research & Therapy*, 24(1), 147. <https://doi.org/10.1186/s13075-022-02834-6>
- VanderWeele, T. J., & Knol, M. J. (2014). A Tutorial on Interaction. *Epidemiologic Methods*, 3(1), 33-72. <https://doi.org/doi:10.1515/em-2013-0005>
- Westerman, K. E., Pham, D. T., Hong, L., Chen, Y., Sevilla-Gonzalez, M., Sung, Y. J., Sun, Y. V., Morrison, A. C., Chen, H., & Manning, A. K. (2021). GEM: scalable and flexible gene-environment interaction analysis in millions of samples. *Bioinformatics*, 37(20), 3514-3520. <https://doi.org/10.1093/bioinformatics/btab223>
- Westerman, K. E., & Sofer, T. (2024). Many roads to a gene-environment interaction. *American Journal of Human Genetics*, 111(4), 626-635. <https://doi.org/10.1016/j.ajhg.2024.03.002>
- Yang, X., Cheng, B., Cheng, S., Liu, L., Pan, C., Meng, P., Li, C. e., Chen, Y., Zhang, J., Zhang, H., Zhang, Z., Wen, Y., Jia, Y., Liu, H., & Zhang, F. (2024). A genome-wide association study identifies candidate genes for sleep disturbances in depressed individuals. *Human Genomics*, 18(1), 51. <https://doi.org/10.1186/s40246-024-00609-5>

RESPONSE TO REVIEWER COMMENTS

[*Blame it on the Sunshine: Genome-wide gene-environment interaction study uncovers 162 novel vitamin D status variants after precise assessment of ambient UVB*]

We thank all the Reviewers for their time and insightful feedback provided. Point-by-point responses to the remarks by Reviewer #1 are provided below.

Reviewer #1 (Remarks to the Author):

Thank you for your feedback. We appreciate the time and effort dedicated to this review. As discussed below, these comments challenged us to think more deeply about our work, and we believe the changes have strengthened our manuscript.

Replication.

In the discussion the authors note “The present study uses data from individuals of European descent, therefore our results are not generalisable to other ethnicities.” However, in their interpretation on what represents ‘a signal’, they do not appear to have considered replication at all, and looking at the results, about half of the signals do not appear to be replicating/ generalisable within the European ancestry populations included in the UK biobank.

The summary statements included in the discussion about ‘replication’ are not helpful for assessing the relevance of the individual signals. Looking at Table S6 and using a statistically very lenient(!) approach where any of the three tests (marginal, joint or interaction) being significant at $p < 0.05$ would represent ‘replication’, for nearly half of the SNP (~150) there was no statistical support. An argument that a sample of ~24,000 would be insufficient in size for this type of lenient replication analysis is not easy to digest, and this does suggest that many of the ‘signals’ may be spurious. What would be interesting, however, is to see which of the variants are replicated and where the evidence appears to be robust. This is where I would have liked to see the paper being focused on and this is where the possible value of this paper would lie.

While I appreciate their effort to replicate (and their note on the difficulties to achieve independent replication in this particular context), it is important to acknowledge that while replication in the European sub-sample of the UK biobank will ensure that a single individual is not included both in the discovery and replication, this does not make it ‘independent’ replication. Biases related to selection and measurement error are likely to be directly correlated between the sample non-British Europeans and white British participants in the UK Biobank, which the least should be mentioned as a limitation.

We appreciate the Reviewer’s feedback and agree that engaging more critically with replication methods and outcomes would add value to the paper. In our efforts to respond, we now add:

- a statistical analysis of replication sample size, and
- replication in two independent cohorts (LURIC from Germany and ORCADES from Scotland).

Firstly, we consider the utility of replication in protecting against false positives. To the same end, we performed extensive quality control of the genetic data (see Zhang et al., 2022 doi.org/10.1093/hmg/ddab203 and Anderson et al., 2010 [nature.com/articles/nprot.2010.116](https://www.nature.com/articles/nprot.2010.116) for discussion on QC and false positives) and sensitivity analyses and applied stringent significance thresholds. Furthermore, crucially, our study replicated the majority of published vitamin D GWAS signals, supporting the validity of our findings. Many of the new GWAS hits (e.g. *PTH*) are strongly aligned with established knowledge, reinforcing their plausibility. Hence, the known SNPs we replicate merit publication and sharing in the field, and SNPs which exhibit a novel signal can remain as putative signals to be further investigated. The conclusion has been revised as follows:

“In summary, using standardised assessments and a precise ambient UVB dose estimate for each participant, we replicated the majority of previously reported 25OHD SNPs, adding further evidence to support these signals. Novel SNPs remain as putative signals that might be

interrogated in future studies through other types of analyses, or in independent GWAS replication when sufficiently large samples become available. These results expand our knowledge of the genetic aetiology of vitamin D status and demonstrate the advantage of the use of accurate environmental exposure data in genetic studies.”

We have edited the quoted statement to align with the robust evidence presented:

“The present study is primarily based on the data from White British individuals living in the UK, therefore our results are not generalisable to other ethnicities.”

With a sample size of $N=338,977$, our study - like many other UK Biobank studies - is exceptionally well powered. However, this strength can become a challenge in the replication phase, as there are currently no cohorts available to us that are on the same scale and with all of the required data, thus increasing the likelihood of false negatives in the replication. The fact that comparable UK Biobank studies do not include replication even when only part of the cohort is used suggests that independent replication is not essential (e.g. Hillary et al., 2024 with 52,363 participants split into discovery and replication [nature.com/articles/s41467-024-51744-5](https://doi.org/10.1093/bioinformatics/btad005); Alcalde-Herrera et al., 2024 split the cohort into 4 discovery groups based on infection or vaccination status, with N up to 240,661, and 4 non-Caucasian validation with N up to 57,851 [nature.com/articles/s41467-024-52890-6](https://doi.org/10.1093/bioinformatics/btad006); or Xu et al., 2025 with only discovery $N = 48,608$ [nature.com/articles/s41467-024-55422-4](https://doi.org/10.1093/bioinformatics/btad004)).

The primary benefit of an independent replication in this case would be to support the *generalisability* of findings, i.e. identifying associations that replicate in ethnicities other than White British. We formally calculated the sample sizes needed for replication using the R package *genpwr* ($p < 0.05$, power 80%). For all the main independent significant SNPs, **Table S6** shows the replication sample size estimates from *genpwr* alongside the results of the replication analysis in three replication cohorts: European UKB ($N = 21,875$), LURIC ($N = 2,909$), and ORCADES ($N = 1,875$). This *genpwr* analysis is described in the supplementary notes and the results are discussed further below.

*“We performed an analysis to determine the sample size required for replication of the variants identified in the discovery cohort, using the R package *genpwr*.⁷ Sample size was estimated for the significant independent SNPs from the marginal model and from the joint model for each variant based on its effect size and minor allele frequency (from the discovery cohort). For the joint SNPs, β_{\circ} effect estimates were used, assuming a true model of additive effects and a 2 degrees of freedom joint test model. For the marginal SNPs, β_{Marginal} effect estimates were used, assuming a true model of additive effects and an additive test model. For both models, the power threshold was set at 80% and the significance as a lenient significance threshold of 0.05 (*genpwr.calc*). We use $p < 0.05$ to estimate the minimum required sample size, noting that more stringent significance thresholds would require larger samples (e.g. the estimated sample size for a variant with $MAF=0.1$ and $\beta=0.1$ is approx. 4300 at $p < 0.05$ and 11,800 at $p < 0.00016$). The estimated sample sizes for a replication p -value of 0.05 ranged from $N = 3,121$ to $N = 958,882$ for the joint test and from 582 to 400,423 for the marginal (**Table S6**). The problem of replication is well-described in the literature wherein larger sample sizes are needed for replication than discovery to achieve enough power, given the constraint of testing for specific variants (see Liu et al.⁸ for an in-depth discussion).”*

Firstly, variants were successfully replicated when our sample size aligned with the required sample size estimated by the *genpwr* analysis. For example, the required sample size was lowest for known variants rs11723621 in *GC* ($N_{\text{genpwr}} = 583$) and rs117913124 in *CYP2R1* ($N_{\text{genpwr}} = 1,282$), which replicated across the three cohorts. Additionally, there were 8 variants with $N_{\text{genpwr}} < 3,500$, all of which replicated in the European subgroup, 5 replicated in LURIC (2 were missing), and 4 in ORCADES (1 missing).

For novel findings specifically: to replicate novel variants associated with genes newly linked with vitamin D, the smallest sample size estimates were for variants in *LINC02548* ($N_{\text{genpwr}} = 43,323$),

C11orf58 ($N_{\text{genpwr}} = 74,883$), and *MOB1B* ($N_{\text{genpwr}} = 77,656$). These estimates illustrate that a replication sample of ~22,000 does not guarantee a sufficiently powered replication of novel hits. Interestingly, these 3 variants were successfully replicated in the European subgroup, likely due to a higher MAF and stronger effects in European vs. White British.

Secondly, genpwr results show that very large sample sizes are needed to replicate most of our novel findings in a GWAS setting, where sample size estimates ranged from 12,131 to 452,020. For some of the variants we find particularly interesting, the estimated sample size is currently prohibitively large, such as in *BMAL1* ($N_{\text{genpwr}} = 281,969$) or *PTH* ($N_{\text{genpwr}} = 109,267$).

Thirdly, we observe that our replication results from the three cohorts are in line with expectations based on the relationship between sample size and effect size/allele frequency. Of 21 SNPs with the strongest effect size ($|\beta_{\text{marginal}}| > 0.1$), several SNPs replicate at Bonferroni-corrected $p < 0.00016$. The p-value for novel SNPs was higher (i.e. less significant) than for known SNPs. This is as expected, since the median effect size and minor allele frequency were lower in novel SNPs ($|\beta_{\text{marginal}}|=0.127$, MAF=0.015) than in known SNPs ($|\beta_{\text{marginal}}|=0.188$, MAF=0.027).

A similar trend was observed for SNPs with effect size between 0.05 and 0.1. Known SNPs that replicated at the genome-wide significance threshold had a higher minor allele frequency (median MAF=0.274), compared to: (i) SNPs that replicated at Bonferroni-corrected $p < 0.00016$ (median MAF=0.061), (ii) those at $p < 0.05$ (median MAF=0.033), and (iii) those that were not statistically significant (median MAF=0.014).

In summary, variants successfully replicated in the context of the replication cohorts' sample sizes. The genpwr analysis suggests that insufficient statistical power may be the underlying reason why some variants did not replicate. Overall, we replicated 152, 33, and 30 out of 308 signals in the three cohorts, respectively (European UKB $N = 21,875$; LURIC $N = 2,909$; and ORCADES $N = 1,875$). This compares favourably to the replication results in the latest vitamin D GWAS (Revez et al., 2020) that reported 8, 17, and 33 SNPs out of 143 independent variants to be significant at $p < 0.05$ in their 3 replication cohorts (QIMR $N=1,632$; UKB subgroup $N=1,632$; SUNLIGHT $N=79,366$). The discussion has been updated as follows:

“The challenge of replication of GWAS results has been discussed at length in the literature.⁶⁸ A statistical power calculation, based on our discovery cohort effect size and MAF, estimated a required median sample size of 103,115 participants to replicate novel variants (Supplementary Notes). As such, given the low frequency and small effect size, we did not have enough power to replicate some of the novel variants, most notably the PTH and BMAL1 variants.”

Given the statistical power issues with replication quantified above, we adopt the approach implemented in previous GWAS, including Revez et al., 2020 mentioned above, focused on evaluating the concordance between effect size estimates. From the marginal model, we observed a significant correlation between the White British and European subgroup ($r=0.96$), the German LURIC cohort ($r=0.76$), and the Scottish ORCADES cohort ($r=0.50$). This compares favourably to the correlation reported in Revez et al., 2020: “pairwise correlations between allele effect size estimates in the different cohorts were all highly significant, ranging from 0.44 between QIMR and UKBR and 0.91 between UKB and SUNLIGHT”. Details of our results are reported in the main text, **Table S6**, and **Fig. S6**.

As suggested by the Reviewer, we increase focus on those findings that do replicate. The following text was reviewed as follows:

“We add novel evidence of interaction in COPB1 and PSMA1, both replicated strongly in the European subgroup and PSMA1 additionally in LURIC.”

“Several variants were identified in genes also implicated in lipid/lipoprotein pathways, including genes previously linked with vitamin D status (e.g. APOE, APOC1, PLA2G3, PCSK9, CELSR2, GALNT2, and CETP) and novel genes, MOB1B and HAVCR1, replicated in the European subgroup and in LURIC, respectively.”

“We replicated 9 previously reported UDP-glucuronosyltransferase genes and identified an additional four (UGT1A1, UGT1A3, UGT2A1, UGT2A2). The novel chromosome 2 variant rs35203651 was nominally significant in the European replication subgroup.”

Finally, we appreciate the Reviewer’s acknowledgement of the challenges of replication in this context and acknowledge the concern arising from the shared selection and measurement bias that may exist in the limitations section:

“Furthermore, replication in the European participants in the UKB cannot be considered independent, because biases related to selection and measurement error are likely correlated to those that may be affecting the discovery sample.”

Supplementary table S6. There is a typo as the authors note that "Significant p-values < 5x10⁻⁸ are shown in bold.", however, many of the bolded p-values are larger than this.

Thank you for spotting this error. The threshold originally applied was 0.00016 (i.e. 0.05/307 the total number of variants tested). Given the added replication results and the updated text, we now show a coloured legend of p-values at thresholds of 5x10⁻⁸, 0.00016, and 0.05.

Some adjustments to interpretation which have been included in the limitations, have not been incorporated into the title or interpretation/summary (end of discussion). Despite acknowledging deficits of their area based UVB measures as measures of actual dosage received by the individual, they claim to have incorporated data on a “precise ambient UVB dose” and “accurate exposure data”. Please revise.

We acknowledge the Reviewer’s suggestion about the use of these terms and for added clarity, we have removed the word ‘*exposure*’ wherever it may lead to misinterpretation, including the title. However, we would respectfully argue that we use terms “precise” and “accurate” appropriately given that we use them in reference to ambient UVB, and not to describe UVB dose received by the participant.

It is important to reiterate that the use of satellite data avoids self-reporting bias, and provides a more objective measure when compared to commonly used UVB proxies, such as self-reported outdoor time or occupation. The Tropospheric Emission Monitoring Internet Service (TEMIS) UVB measurements have been evaluated through several validation studies comparing TEMIS data with ground-based observations, and these robustly demonstrated their precision. We have checked all occurrences of “precise”, “accurate” and “refined” in text and confirm these references are appropriately used to describe ambient/environmental UV. The limitations we describe occur downstream, as we do not know the actual dose received by participants living at the location with the measured level of ambient radiation. This is discussed as one of the limitations:

“Finally, while CW-D-UVB accurately captures the environmental availability of UVB, the actual dose received by each participant will vary depending on clothing, when and how much time they spend outdoors, and other personal and behavioural factors. A more accurate measure of the dose received may be possible with future advances in the available tools, such as scalable dosimeters or linked phone GPS indoor/outdoor data, that would further improve assessment of actual UVB exposure.”